# Recent trend reversal for declining European seagrass meadows

Carmen B. de los Santos [1], Dorte Krause-Jensen [2,3], Teresa Alcoverro[4], Núria Marbà [5], Carlos M. Duarte [6], Marieke M. van Katwijk [7], Marta Pérez[8], Javier Romero [8], José L. Sánchez-Lizaso [9], Guillem Roca[5], Emilia Jankowska[10], José Lucas Pérez-Lloréns[11], Jérôme Fournier[12], Monica Montefalcone[13], Gérard Pergent[14], Juan M. Ruiz[15], Susana Cabaço[1], Kevan Cook[16], Robert J. Wilkes[17], Frithjof E. Moy[18], Gregori Muñoz-Ramos Trayter[19], Xavier Seglar Arañó[19], Dick J. de Jong[20], Yolanda Fernández-Torquemada [9], Isabelle Auby[21], Juan J. Vergara [11] & Rui Santos [1]

Seagrass meadows, key ecosystems supporting fisheries, carbon sequestration and coastal protection, are globally threatened. In Europe, loss and recovery of seagrasses are reported, but the changes in extent and density at the continental scale remain unclear. Here we collate assessments of changes from 1869 to 2016 and show that 1/3 of European seagrass area was lost due to disease, deteriorated water quality, and coastal development, with losses peaking in the 1970s and 1980s. Since then, loss rates slowed down for most of the species and fast-growing species recovered in some locations, making the net rate of change in seagrass area experience a reversal in the 2000s, while density metrics improved or remained stable in most sites. Our results demonstrate that decline is not the generalised state among seagrasses nowadays in Europe, in contrast with global assessments, and that deceleration and reversal of declining trends is possible, expectingly bringing back the services they provide.

[1] Centre of Marine Sciences of Algarve (CCMAR), University of Algarve, Gambelas, 8005-139 Faro, Portugal. [2] Department of Bioscience, Aarhus University, Vejlsøvej 25, 8600 Silkeborg, Denmark. [3] Arctic Research Centre, Department of Bioscience, Aarhus University, Ny Munkegade 114, Building 1540, 8000 Århus C, Denmark. [4] Centre d'Estudis Avançats de Blanes (CEAB-CSIC), Carretera Acc, Cala Sant Francesc 14, 17300 Blanes, Girona, Spain. [5] Global Change Research Group, Institut Mediterrani d'Estudis Avançats (IMEDEA, CSIC-UIB), Miquel Marquès 21, 07190 Esporles, Illes Balears, Spain. [6] King Abdullah University of Science and Technology (KAUST), Red Sea Researh Center (RSRC) and Computational Bioscience Research Center (CBRC), Thuwal 23955-6900, Saudi Arabia. [7] Department of Environmental Science, Institute for Water and Wetland Research, Radboud University Nijmegen, Heyendaalseweg 135, 6525 AJ Nijmegen, The Netherlands. [8] Department of Evolutionary Biology, Ecology, and Environmental Sciences, University of Barcelona, Av. Diagonal 643, 08028 Barcelona, Spain. [9] Department of Marine Sciences and Applied Biology, University of Alicante, PO BOX 99, 03080 Alicante, Spain. [10] Institute of Oceanology, Polish Academy of Sciences, Powstańców Warszawy 55, 81-712 Sopot, Poland. [11] Department of Biology, Faculty of Marine and Environmental Sciences, Marine Research Institute, University of Cádiz, 11510 Puerto Real, Cádiz, Spain. [12] Muséum National d'Histoire Naturelle, CNRS UMR 7204 Centre d'Ecologie et des Sciences de la Conservation, Station de Biologie Marine, Place de la Croix BP225, 29182 Concarneau Cedex, France. [13] DiSTAV, Department of Earth, Environment and Life Sciences, University of Genoa, Corso Europa 26, 16132 Genoa, Italy. [14] Coastal Ecosystem Team (FRES 3041/UMR 6134), University of Corsica, BP 52, 20250 Corte, France. [15] Seagrass Ecology Group, Oceanographic Center of Murcia, Spanish Institute of Oceanography, C/ Varadero, 30740 San Pedro del Pinatar, Murcia, Spain. [16] Natural England, Pydar House, Truro TR1 1XU, UK. [17] Environmental Protection Agency, John Moore Road, Castlebar F23 KT91 Co. Mayo, Ireland. [18] Institute of Marine Research, P.O.Box 1870 Nordnes, 5817 Bergen, Norway. [19] Medi Ambient, Ajuntament de Badalona, Plaça de la Vila 1, 08911 Badalona, Barcelona, Spain. [20] Department Sea and Delta, Ministry of Infrastructure and the Environment, Rijkswaterstaat, 4330 KA Middelburg, The Netherlands. [21] LER Arcachon-Anglet, IFREMER, Quai du commandant Silhouette, 33120 Arcachon, France. Correspondence and requests for materials should be addressed to C.B.d.l.S. (email: cbsantos@ualg.pt)

Seagrasses, marine flowering plants forming underwater meadows, play a key global role in supporting fisheries production[1], climate change mitigation[2], and coastal protection[3]. However, they rank among the most threatened ecosystems on Earth, with global loss rates accelerating from 0.9% $yr^{-1}$ in the 1940s to 7% $yr^{-1}$ toward the end of the 20th century[4]. Rapid global losses are largely attributable to anthropogenic impacts, mainly loss of water quality and coastal development[5] and, more recently, to extreme events, such as storms[6] and marine heat waves[7,8]. The loss and deterioration of seagrass meadows are compromising the important services they provide, so the call for their global conservation is gathering momentum in order to secure their future[9,10]. Yet, seagrass conservation implies many challenges[11], especially because seagrasses lack the charisma of other coastal ecosystems[12]. One of the challenges in seagrass conservation is informing on their status and condition[11], since spatial and temporal data on seagrass extent and density are normally scattered or scarce in most regions, as well as disparate in terms of the metrics of change assessed.

In Europe, either loss[13–17] and recovery or stability[18–21] are reported for seagrass meadows, with an overall trend toward decline reported for the Mediterranean endemic species *Posidonia oceanica*, which has lost between 13% and 50% of its areal extent since 1960[16,17]. Along with *P. oceanica*, three other native seagrass species occur along European coasts: *Zostera marina*, *Z. noltei*, and *Cymodocea nodosa*. However, a comprehensive evaluation of seagrass status across Europe is still lacking, and thus the magnitude and direction of changes in seagrass extent and density at the continental scale remains unclear, in particular for species other than *P. oceanica*. The identification of the causes of seagrass loss and recovery is equally important to implement effective management actions to halt current losses and boost their recovery. Europe is a distinctive geographical region for having adopted in 2000 the seagrasses as sensitive quality elements providing a diagnostic of ecosystem health under the European Union (EU) Water Framework Directive (WFD)[22–26], which aims at maintaining good ecological status in European waters. The focus on seagrasses as indicators of ecosystem health of coastal waters has led to increasing monitoring efforts across the European continent in the past two decades[16,24]. This monitoring effort, along with older data on seagrass extent in locations with a long history in seagrass monitoring and mapping, provides the basis to examine seagrass trajectories at the continental scale.

Here we assess the continental-scale trends of the extent and density of European seagrasses and identify the causes of change, both for loss and recovery. We rely on a compilation of assessments of change in 737 seagrass sites along the coasts of 25 European countries, from 1869 to 2016, including occurrence, area extent, depth limits, cover, shoot density, and biomass. We assess the pattern of the trajectories of increase, decline, and no change for the entire time period and find a prevalence of declines, with about one third of the seagrass area lost. We additionally investigate the trends of the rates of change and the evolution of the trajectories from the 1950s to the 2000s, and we find a slowdown in the losses and an improvement of the trajectories in the 1990s and 2000s, resulting in a reversal of the declining trends. We conclude that decline is not the generalized state among seagrasses nowadays in Europe.

## Results

### Overall changes of European seagrasses from 1869 to 2016.
Across the observational record (1869–2016), and integrating all the metrics of change, there was a prevalence of sites reporting decline (49%) compared to those reporting increase (22%) or showing no change (29%), when accounting for all the species

(Fig. 1). The highest proportion of declines, integrating extent and density metrics, was reported for *Z. marina* and *C. nodosa*, whereas the lowest was for *P. oceanica* (Fig. 1). The extent metrics (presence, area extent, depth limits; Supplementary Fig. 1) declined in 68% of seagrass sites, whereas density metrics (cover, shoot density, biomass; Supplementary Fig. 1) declined only in 31% of sites. Declines were reported mainly for seagrass area and depth limits (Fig. 2a), and in all cases, the mean specific rate of change of declining sites was higher than of increasing sites (Fig. 2b).

The area losses and gains of European seagrasses added up to 40,411 and 4,727 ha, respectively, resulting in a net loss of 35,684 ha between 1869 and 2016 (Table 1), which represents 29% of the maximum area documented in our compilation (122,582 ha). The species experiencing the largest area declines were *Z. marina* and *C. nodosa*, with net losses representing 57% and 46% of their maximum area assessed, respectively (Table 1). Area gains were higher for *Z. noltei* and *C. nodosa*, representing 8.1% and 15.6% of their total assessed area, respectively. In contrast, *P. oceanica* and *Z. marina* gains represented a small percentage of the maximum area compiled (0.5% and 2.1%, respectively).

Seagrass meadows in the Baltic Sea experienced the largest percentage of losses of area (67%), followed by the Atlantic (36%) and Mediterranean (21%) coasts (Table 1). Among the 212 seagrass sites reporting changes in area, 16 sites accounted for 75% of the losses, whereas 5 sites accounted for the same percentage of gains. The largest net losses in seagrass areas for single locations were registered for *Z. marina* (7190 ha in the Wadden Sea, 3296 ha in Puck Bay, 2490 ha in Odense Fjord, 2422 ha in Golfe du Morbihan, 1358 ha in Lake Grevelingen) and *P. oceanica* (4364 ha, Cape Circeo and Sperlonga, Italy), whereas the largest net gains were documented for *Z. noltei* (2434 ha in the Northfrisian Wadden Sea and 378 ha in Bourgneuf Bay), *C. nodosa* (304 ha, Alfacs Bay), and *Z. marina* (210 ha in Rance-Fresnaye, 180 ha in Baie de Morlaix, and 128 ha in Les Abers Large, France; 199 ha in Mariager Fjord, Denmark). The magnitude of the losses in seagrass area (median = 46 ha, mean ± s.e.m. = 297 ± 72 ha, $n = 136$) was significantly larger than for area gains (median = 12 ha, mean ± s.e.m. = 84 ± 44 ha, $n = 56$; two-sample Kolmogorov–Smirnov test, $D = 0.29$, $p = 0.0018$, Supplementary Fig. 2).

### Decadal changes of European seagrasses from 1950s to 2000s.
The decadal rate of area loss of European seagrasses revealed an acceleration over the second half of the 20th century to peak at −33.6% $decade^{-1}$ in the 1970s (Fig. 3). Losses subsequently slowed down to lower rates in the 1980s (−27.0% $decade^{-1}$), in the 1990s (−16.1% $decade^{-1}$) and in the 2000s (−8.3% $decade^{-1}$), while the decadal rate of area gains increased during the 1990s and 2000s. Consequently, for the first time since the 1950s, a large net gain in area was attained in the 2000s at a specific rate of 20% $decade^{-1}$. The large increase in area observed during the 2000s was mainly attributed to *Z. noltei* gains in the Atlantic coasts (79% of total gains, 15 sites) and in the Mediterranean Sea (7.2%, 1 site), and, secondarily, to *Z. marina* in the Atlantic coast (9%, 11 sites) and in the Baltic Sea (2.1%, 3 sites) (Supplementary Fig. 3). The species-specific decadal rate of change in area revealed that the trend reversal detected in the 2000s was due to the slowdown of losses of all the species (except for *C. nodosa*, losses of which overpassed gains in the 2000s), along with the fast recovery of the *Zostera* spp. in the 2000s (Supplementary Fig. 3). The mean specific rates of change of area, density metrics (biomass, cover, and shoot density), and depth limits (upper and lower) were not significantly different from 0 in the 2000s (Wilcoxon signed-rank test, $V = 1745$, $p = 0.64$; $V = 1887$, $p = 0.25$; and $V = 37$, $p = 0.12$, respectively; Fig. 4a–c). From 1950s to

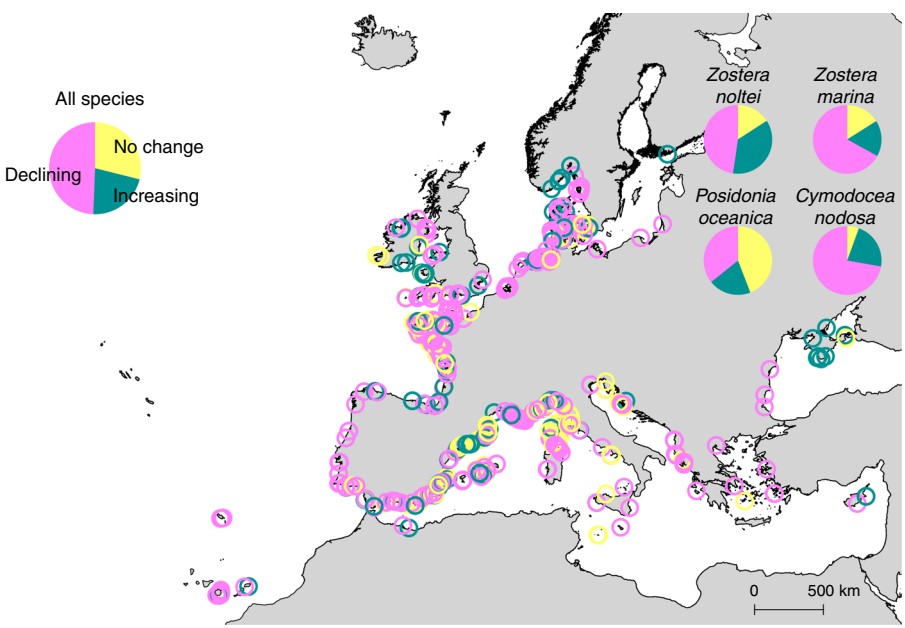

**Fig. 1** Distribution of compiled seagrass sites in Europe and their trajectories. Seagrass sites in Europe showing no change (yellow circles, $n = 213$), increase (green circles, $n = 160$), and decline (magenta circles, $n = 364$) trajectories based on the available time series reports between 1869 and 2016, thus corresponding to different time windows. Pie charts show the overall and species-specific frequencies of trajectories. Number of sites showing decline, increase, and no change trajectories are, respectively: 128, 72, and 158 for *Posidonia oceanica*; 146, 37, and 35 for *Zostera marina*; 39, 12, and 3 for *Cymodocea nodosa*; and 51, 39, and 17 for *Zostera noltei*

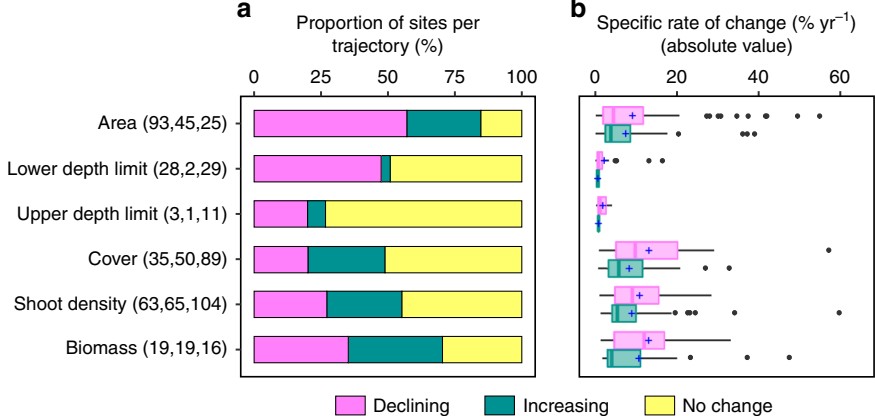

**Fig. 2** Overall changes of European seagrasses per metric of change. **a** Percentage of seagrass sites showing decline, increase, and no change for each metric, and **b** specific rates of change of declining and increasing sites for each metric (% yr$^{-1}$, in absolute value). Number of sites showing decline, increase, and no change trajectories are given between brackets. In the boxplots, the plus (+) represents the mean, the solid line the median, the lower and upper hinges are the first and third quartile, whiskers extends to the largest and smallest values no further than 1.5×inter-quartile range, and data beyond the end of the whiskers are plotted as points. Source data are provided as a Source Data file

**Table 1 Gains and losses in area of European seagrasses per species and regions**

| Category | Maximum compiled area, ha (*N*) | % Area lost (*N*) | % Area gained (*N*) | Total net change, ha (*N*) |
|---|---|---|---|---|
| Species | | | | |
| *Posidonia oceanica* | 38,420 (57) | 19 (37) | 0.5 (6) | – 6990 (43) |
| *Zostera marina* | 40,770 (60) | 57 (53) | 2.1 (13) | – 22,206 (66) |
| *Cymodocea nodosa* | 2,320 (23) | 46 (17) | 15.6 (4) | – 710 (21) |
| *Zostera noltei* | 41,072 (72) | 22 (29) | 8.1 (33) | – 5779 (62) |
| Regions | | | | |
| Mediterranean Sea | 46,854 (78) | 21 (52) | 1.2 (11) | – 9388 (63) |
| Atlantic Ocean | 65,471 (120) | 36 (75) | 6.0 (40) | – 19,696 (115) |
| Baltic Sea | 10,256 (14) | 67 (9) | 2.5 (5) | – 6600 (14) |
| Total | 122,582 (212) | 33 (136) | 3.9 (56) | −35,684 (192) |

*N* number of seagrass sites based on the available time series between 1869 and 2016

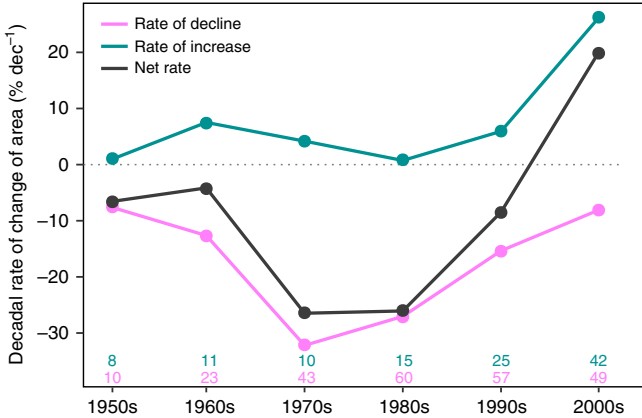

**Fig. 3** Decadal rate of change of area of European seagrasses (1950s–2000s). Decadal analysis includes time series >8 years. Number of sites per decade and trajectory are given at the bottom of the plot

1990s, the number of sites, within each decade, experiencing an improvement in the area trajectory was always lower than those worsening, but this pattern changed in 2000s, when the number of sites improving surpassed the number of those worsening (Fig. 4d). In terms of density metrics and depth limits, there was an increase of sites improving and in steady state (not improvement, not worsening) from the 1980s to the 2000s, even though there was also an increase of sites getting worst (Fig. 4e, f).

**Causes of change in European seagrasses**. The causes for decline or increase of European seagrass meadows were reported for 31% and 7% of the compiled seagrass sites, respectively. Most of the declines were attributed to water quality degradation (26%) and wasting disease (25%), followed by coastal modification (16%), mechanical damage (14%), and multiple causes (12%, Fig. 5a). Whereas the wasting disease caused by *Labyrinthula* sp. was the dominant driver for losses of *Z. marina*, losses of other seagrass species were dominated by water quality degradation (*P. oceanica* and *Z. noltei*) and coastal modification (*C. nodosa*) (Fig. 5a). Seagrass recovery was mostly (68%) attributed to management actions (Fig. 5a), which included improvement of water quality, reduction of industrial sewage, and anchoring and trawling regulation. The rest of cases were attributed to natural colonization that could not be directly associated with any human intervention (Fig. 5a), which included the recovery from wasting disease in the 1950s, recovery after drastic losses in coastal lagoons caused by floods. Water quality degradation was the major loss factor in the 1970s, whereas losses due to extreme events became the most important cause of decline during the 2000s (Fig. 5b, c). Management intervention along with natural colonization emerged as drivers of recovery in the 1990s and 2000s (Fig. 5d).

## Discussion

From 1869 to 2016, about one third of the area of European seagrasses was lost in relation to the maximum compiled area, due to several causes including wasting disease, water quality degradation, coastal development, mechanical disturbance, and the combination of them. However, and contrary to other global reports on seagrass losses[4,5], this work reveals for the first time since the 1950s a trend reversal for declining European seagrass meadows at the end of the 20th century that continued through the 2000s. Whereas losses occurred in all regional seas and species, seagrass gains were concentrated in fewer locations and were mostly due to the recovery of *Zostera* species.

The predominant seagrass trajectory of our compilation was decline, revealed mostly by area and depth limit changes rather than density metrics. This does not mean that those are the best indicators of seagrass loss but rather the consequence of the fact that area and depth limits have been reported more often, since the beginning of seagrass studies in Europe. Density metrics were mainly introduced in the past decades after the highest seagrass losses of the 1970s had occurred, mostly in 2000s because of the broad geographical monitoring imposed by the WFD. Loss of seagrass area was mostly attributed to the species *Z. marina* and *C. nodosa*. The wasting disease outbreak during the 1930s decimated large pristine *Z. marina* areas along the Atlantic coast[27–29]. Other causes behind the seagrass losses in Europe included water quality degradation, coastal modification, and mechanical impacts, in accordance to those previously identified at the global scale[4,5]. Seagrass declines in Europe during the 20th century were reported elsewhere for *P. oceanica*[16,17], with an estimated area loss of 13–50%, and for *Z. marina* in Nordic countries[30]. For *Z. noltei* and *C. nodosa*, this is the first assessment revealing both losses and gains at the continental scale.

Loss rates of European seagrasses peaked in the 1970s and 1980s and started to slowdown in magnitude toward the end of century, when it reached the loss rate of the 1950s (Fig. 3). Decadal rates before the 1950s were not possible to assess due to the data deficiency. The combination of this deceleration with the area gains observed during the 1990s and 2000s, mainly due to large expansions of *Z. noltei* and *Z. marina* along the Atlantic coasts (79% and 9% of total gains, respectively), led to the recent reversal of the negative decadal rate of net change during the 2000s. The improvement of the seagrass trajectory in Europe was also evident in density metrics, which become stable or improved during the 1990s and 2000s. Most of the sites reporting gains in seagrass area during the 2000s did not include the causes for those gains. The available information indicates that the largest increase in seagrass area during the 2000s, of *Z. noltei* in the Northern Wadden Sea (9017 ha), was due to the reduction of nutrient loads[19,31]. The second biggest contribution was the recovery of 913 ha of *Z. noltei* at the Vaccarès lagoon, France, due to the natural restoration of water clarity and salinity, which had been drastically reduced by two consecutive river floods[32]. Another contribution to area gains, the recovery of *Z. marina* in Puck Bay, Poland, was ascribed to an improvement in water quality following management actions to reduce water pollution[33]. Thus a combination of natural recovery of seagrasses after environmental improvement related or not to management actions may explain the recent positive trajectory of European seagrasses.

The effects of management actions to improve water quality on seagrass recovery are well documented at the national and subnational scales: nutrient input reduction to fjords in Denmark resulted in an increase of the depth limit of eelgrass[34], decreased nitrogen inputs in a Portuguese estuary in 1998 reversed the declining trajectory of *Z. noltei* after severe eutrophication events during the 1980s and early 1990s[35], and the increase of wastewater treatment plants from 2003 to 2010 along the Catalonia coasts in Spain resulted in significant improvements of water quality and of the biochemical indicators of *P. oceanica*[36]. These cases add up to success stories reported outside of Europe, such as the recovery of seagrasses in Chesapeake Bay[37], Tampa Bay[38], and in Mumford Cove[39], following water quality improvement.

Even though the structure of the data compiled here do not allow us to relate European seagrass recovery to specific management actions, seagrass meadows in Europe may have benefited from policies and management initiatives adopted in the 1990s to reduce nutrient loading from urban waters[40] and from agricultural sources[41]. The subsequent identification of seagrasses as key

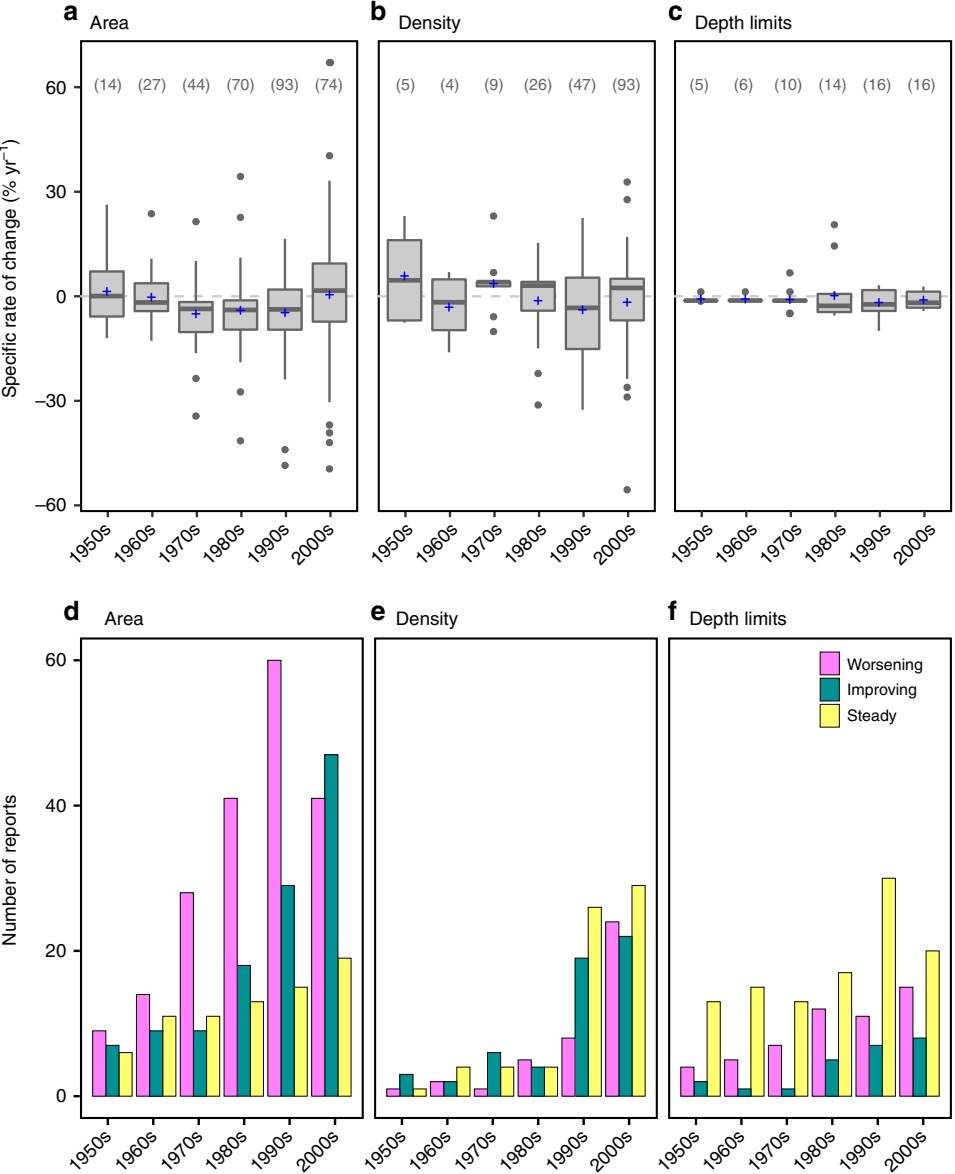

**Fig. 4** Specific rates of change and trajectory evolution of European seagrasses (1950s–2000s). Specific rate of change (% yr$^{-1}$) integrating sites showing either increase or decline trajectories in **a** area, **b** density, and **c** depth limits and number of sites where the trajectory improved, worsened, or stayed steady in **d** area, **e** density, and **f** depth limits. Decadal analysis includes time series >8 years. Number of reports per decade in **a–c** is given between brackets. In the boxplots, the plus (+) represents the mean, the solid line the median, the lower and upper hinges are the first and third quartile, whiskers extends to the largest and smallest values no further than 1.5×inter-quartile range, and data beyond the end of the whiskers are plotted as points

indicators of ecosystem health[22–26] within the EU WFD[42] in the 2000s brought seagrass onto the attention of managers and made it mandatory that seagrass meadows in moderate or poor ecological status were restored to reach good ecological status before 2015, with steep monetary penalties in case of default. The designation of marine protected areas in European countries, either by national policies or under the umbrella of the EU Habitat Directive adopted in 1992[43], may have also contributed to the reported slowdown of the seagrass losses in the past decades. The Habitats Directive, which ensures the conservation of rare, threatened, or endemic species and characteristic habitat types, including *Z. marina* and *P. oceanica*, led to the creation of a network of conservation sites, the Natura 2000[44]. This network greatly increased the number of seagrass protected sites across Europe, which included 322 sites with *P. oceanica* meadows in the Mediterranean in 2006, encompassing a total of 2700 km[2][45]. The Habitats Directive and the

WFD may have been relevant, complementary tools for seagrass conservation in Europe, combining direct habitat protection with water quality restoration, respectively. The Habitats Directive particularly benefits *P. oceanica*, a species resistant to disturbances but with an extremely low recovery capacity[46], whereas the WFD particularly benefits the other species with a high recovery capacity once water quality is improved.

In conclusion, the results presented here show serious declining trajectories of European seagrasses since 1869 but revealed a recent trend reversal in seagrass extent and density. This improvement is likely related to actions to conserve and restore seagrass meadows in Europe by reducing nutrient loading, improving water quality, or by direct habitat protection. The recovery of seagrass meadows in Europe, especially those with a fast growth capacity, brings the expectation of the return of services and benefits they provide.

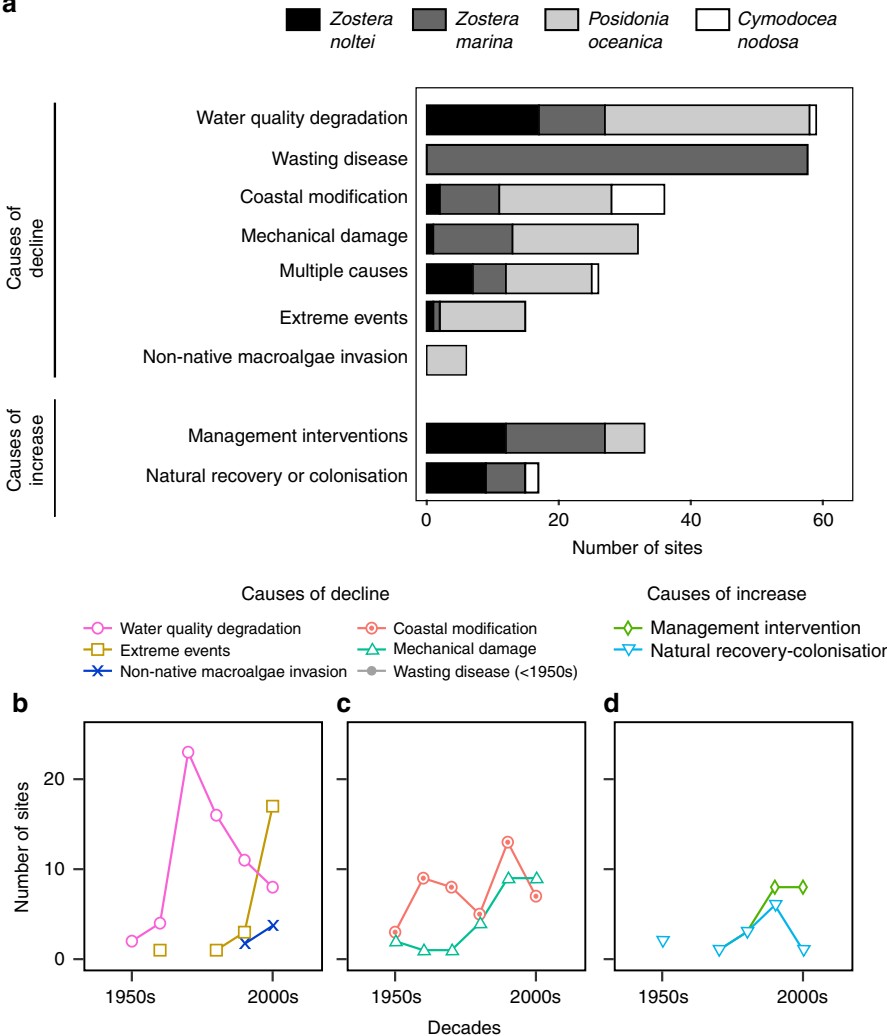

**Fig. 5** Causes of change in European seagrasses. **a** Number of sites showing decline or increase due to different causes and decadal timeline of the number of sites that reported declines (**b–c**) and increases (**d**) from 1950s to 2000s. Only non-zero data are shown and lines connect only consecutive decades

## Methods

**Data compilation**. We compiled the available assessments of the seagrass species *C. nodosa*, *P. oceanica*, *Z. marina*, and *Z. noltei* along European shores. The Lessepsian migrant species *Halophila stipulacea*, present across the Eastern and Central Mediterranean[47], was excluded owing to the scarce number of time series available. Published sources were gathered using the search browser GoogleScholar in April 2013 and then again in December 2017, by combining keywords related to seagrasses (seagrass, *Posidonia*, *Cymodocea*, *Zostera*, or eelgrass), with changes (loss, gain, decline, increase, stability, recovery, change, long-term, evolution, dynamic, impact, and diachronic), and names of European countries or regional seas. These searches, together with the authors' personal data collections, yielded 520 potential time-series sources, from which 241 were kept. The reasons for exclusion were: (1) inaccessible source, (2) source including a site already compiled or updated in another more recent source, (3) source being a review or compilation (in those cases, the source was consulted to find potential studies to assess), and (4) sources in which data criteria were not met (e.g., type of metrics). Sources were sorted in 166 journal articles, 33 technical reports, 12 book chapters, 13 conference proceedings, and 6 PhD or MSc theses. In addition, 11 verified databases were facilitated by participants of the COST Action ES0906 (Seagrass productivity: from genes to ecosystem management) in 2013. The thirty-four European sites included in the previous global review of seagrass trends[4] (16% in their database) were added to our database, accounting for 4% of our compilation, and 13 of them were updated based on new studies reporting recent observations. Several data verification steps were conducted, including independent checks by authors of the sources they provided, proof-reading the data twice, and identifying and verifying outliers against original sources. When not directly available from the source, data values were inferred from graphical representations using GraphicClick (©2008

Arizona Software) or ImageJ (US National Institutes of Health). Dataset and references are available from Supplementary Data 1.

The final compilation included 1042 assessments of change for 737 seagrass sites: 56% of sites were in the Mediterranean Sea (including the Black Sea), 38% in the European North Atlantic Ocean, and 6% in the Baltic Sea, across 25 countries (EU-countries: Bulgaria, Croatia, Cyprus, Denmark, Finland, France, Germany, Greece, Ireland, Italy, Lithuania, Malta, Poland, Portugal, Romania, Slovenia, Spain, Sweden, The Netherlands, United Kingdom, and non-EU countries: Albania, Monaco, Norway, Turkey-in-Europe, Ukraine). The overall dataset covered 147 years, from 1869 to 2016, with the observation effort increasing exponentially over time. The duration of the time windows were highly variable among sites, from 1 to 121 years with a median of 9 years. We retrieved information for 358 sites of *P. oceanica* (49%), 218 of *Z. marina* (30%), 107 of *Z. noltei* (15%), and 54 of *C. nodosa* (7%).

Each site was classified by the seagrass species for monospecific meadows or by the dominant species for mixed meadows. When a site included separate assessments for depth (i.e., shallow meadow and deep meadow) or for co-occurring species, those observations were considered as independent sites. Some sites (239) lacked information on the reference year (e.g., disappearance, colonization, drastic reduction in area or density, loss of area due to bottom trawling) or only included presence data, so they were only considered for the overall analysis of number of sites by trajectory (Fig. 1, Supplementary Fig. 1). When different studies covering the same site overlapped in time, the longest assessment was selected. If two different sources covered the same site and/or time-window but used different metrics, both of them were included in the dataset. The metrics compiled were classified as seagrass extent metrics [presence/absence, area (ha), and depth limits (m)], seagrass density metrics [cover (%), shoot density (shoots m$^{-2}$), and total or

above-ground biomass density (g dry weight m$^{-2}$)]. Of the 737 seagrass sites compiled, 48% included extent assessments, 43% density assessments, and 9% included both.

**Overall and decadal analyses**. The trajectories of seagrass extent and density were investigated irrespective of the time window of the observation (overall analysis). Trajectories were classified as increasing, declining, or no change based on the percentage of change between the final and initial values of metrics and the presence/absence data. When both depth limits were available or when more than one density metric was reported, the average percentage of change was used. The criteria to consider that there was a relevant change was set at 10% for extent metrics (which is typically within the error of area and depth limit assessment techniques[4]) and 25% for density metrics (which is the average coefficient of variation of density metrics of the dataset). For example, if a final density observation was within the 75–125% of the initial observation, the trajectory was defined as no change; otherwise, if the final value was <75% of the initial, it was defined as declining, and if it was >125%, it was defined as increasing. When a combination of extent and density metrics was available for the same site, there were three possible outcomes for the final trajectory: (1) the metrics indicated the same trajectory, (2) some metrics showed no change and others showed an increasing or declining trajectory, and (3) metrics showed opposite trajectories. In the first case, the final trajectory was given by the unique trajectory; in the second case, it was given by the trajectory showing an increase or decline (e.g., area showed no change and density increased, the trajectory attributed to the site was increasing); in the third case, the decision was based on the following hierarchy: area > depth limits > density (e.g., area declined and density increased, final trajectory was declining). For each seagrass site, the specific rate of change (% yr$^{-1}$) of metrics was calculated as $100 \times (\ln(V_f/V_i)/t)$, where $V_i$ and $V_f$ are the initial and final observations, respectively, and $t$ is the observational time interval (yr).

Time series with at least 8 years (43% of total) were used to evaluate the decadal trends of seagrass metrics. For each site, the specific rate of change (% yr$^{-1}$) and trajectory for each metric and decade were calculated using decade's boundaries that were, when necessary, interpolated using the specific rate of change between the two observations closest to the decade boundaries (Supplementary Data 2). The decadal trajectories were sorted in increasing, declining, or no change using the same criteria as in the overall analysis. Then the evolution of the trajectory over two consecutive decades in a specific site was defined as improving (from no change to increasing, from declining to no change or increasing, or from increasing to increasing), worsening (from no change or increasing to declining, or from declining to declining), or steady (when the trajectory is no change from increasing or no change) to assess the changes in the trajectory over time (Fig. 4d–f). The decadal rate of change of area (% decade$^{-1}$), for all the species (Fig. 3) and each of them (Supplementary Fig. 3), was computed from the sum of areas of sites with area documented at the start of two consecutive decades. The decadal descriptors were calculated only from 1950s onwards due to the limited sample size in previous decades (<20 sites). The 2010s decade was incomplete and thus excluded.

**Causes of change**. The causes of decline in seagrass extent and/or density were identified in the compiled sources and classified as: (1) coastal modification (including harbor construction, dredging, beach filling, land reclamation, construction of pipeline, dams and breakwaters, river diversion and storm drains, relocation inlet and other coastal works), (2) water quality degradation (including input of nutrients and organic matter from fish and shellfish farming and urban sewage, macroalgae overgrowth due to eutrophication, general water degradation, industrial pollution, brine sewage, marine sewage, general pollution), (3) mechanical damage (including bottom trawling, anchoring and mooring, clam digging and bait collection, seagrass harvesting, culture farm occupation, explosives, and other local human activities), (4) non-native macroalgae effects, (5) wasting disease, (6) overgrazing (including sea urchins, waterfowl, and others), (7) extreme events (including heat waves, storms and heavy rainfalls, flood events), and (8) multiple causes, i.e., a combination of two or more causes of decline (Supplementary Table 1). Two categories were defined for causes of increase: (1) natural recovery or colonization, (2) management intervention (positive changes due to regulation and management, including removal/reduction of direct impacts such as improvement of water quality, trawling regulation, reduction of industrial sewage, anchoring regulation, and others). Restoration was not among the causes of seagrass gain in the compiled sources. A decadal analysis was done to investigate the evolution of the frequency of the causes of change (decline or increase) from the 1950s to the 2000s (when a site had multiple causes assigned, that site was counted for every cause). When a site showed a change in the trajectory over time, the cause of the change and change in area, if known, were attributed to each trajectory (e.g., decline of 40 ha during the 1980s due to eutrophication, followed by an increase of 2 ha during the 1990s due to management actions).

**Statistical analysis**. The distributions of the increasing and decreasing seagrass areas per site were compared by a two-sample two-sided Kolmogorov–Smirnov test. Statistical difference from zero in the specific rates of change of area, depth limits, and density in the 2000s were assessed by two-sided Wilcoxon signed-rank test, after checking that data did not meet parametric assumptions. Statistical significance was set at $\alpha = 0.05$. Data and statistical analysis were performed using R Statistical Package[48].

**Reporting summary**. Further information on research design is available in the Nature Research Reporting Summary linked to this article.

## Data availability
The dataset of compiled seagrass sites and time series are available as Supplementary Data 1 and Data 2, respectively. Source data of tables and figures are provided as a Source Data file. All other relevant data are available on request.

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

## Acknowledgements

This research was supported by the COST Action ES0906. C.B.d.l.S. acknowledges the support by the Short-Term Scientific Missions (COST-STSM-ES0906-06377, -06441 and -06419) and the Portuguese Foundation for Science and Technology (FCT) fellowship (SFRH/BPD/119344/2016). C.B.d.l.S. and R.S. acknowledge the support of FCT through project UID/Multi/04326/2019. D.K.-J. was supported by the European Union (WISER 226273), the Danish Strategic Science Foundation (NOVAGRASS, 0603-00003DSF), and the Ministry of Environment and Food of Denmark (33010-NIFA-16-651). N.M. was supported by Spanish Ministry of Economy, Industry and Competitiveness (MedShift, CGL2015-71809-P), by the European Union (M&M's, EVK3-CT-2000–00044 and WISER 226273), by the Fundación BBVA (Proyecto Praderas), and the Spanish Government (MEDEIG. CTM2009–07013). King Abdullah University of Science and Technology (KAUST) supported C.M.D. through baseline funding. J.M.R. acknowledges the project Monitoring network of *Posidonia oceanica* meadows and global climate change of the Murcia Region funded by the Autonomous Government of the Murcia Region (General Directorate of Livestock and Fishery), the European Fishery Fund (EFF 2007–2013), and European Maritime and Fisheries Fund (EMFF 2014–2020), and the Spanish Ministry of Industry, Economy and Competitiveness (UMBRAL, CTM2017-86695-C3-2-R).

## Author contributions

C.B.d.l.S., D.K.-J., T.A., N.M., M.M.v.K., M.P., J.R., J.L.S.-L. and R.S. designed the study; C.B.d.l.S. compiled and harmonized the data; C.B.d.l.S., D.K.-J., T.A., N.M., M.M.v.K., C.M.D., and R.S. analyzed data; C.B.d.l.S. wrote the manuscript with support from C.M.D., D.K.-J., T.A., N.M., M.M.v.K and R.S. D.K.-J, T.A., N.M., C.M.D., M.M.v.K., M.P., J.R., J.L.S.-L., G.R., E.J., J.L.P.-L., J.F., M.M., G.P., J.M.R., S.C., K.C., R.J.W., F.E.M., G.M.-R.T., X.S.A., D.J.d.J., Y.F.-T., I.A., J.J.V. and R.S. contributed with significant datasets and read and approved the submitted version of the manuscript.

## Additional information

**Competing interests:** The authors declare no competing interests.

