## [Peer Review File · Nature Communications]

Reviewers' comments:

Reviewer #1 (Remarks to the Author):

In this MS, de los Santos and colleagues review changes in seagrass cover in European waters over the last century and a half. The data shows that over the last 147 years there's been a loss of about a third of seagrass cover. The authors identify a period of peak loss in the 1970s- 1980s, as well as a reversal in the overall trend of decline in the 2000s, when seagrass recovery rates become faster than seagrass loss. All of this is complemented with data on the causes leading to both declines and recovery.

This is an exciting and timely MS that should be of great relevance to conservation biologists broadly. The exceptional long temporal span and broad spatial scale provides a unique and highly valuable perspective that invites some optimism with regards to the future state of seagrass meadows, as the data demonstrates that many of the recent management interventions to stop seagrass decline are working effectively. This paper is likely to become highly influential among both conservationists and marine ecologists.

I have minor comments/ recommendations that I think would strengthen the MS, which I detail below:

1. L69 Abstract and L223-L224 – Statement that recovery of seagrass is bringing back services and benefits provided by seagrasses is not supported by the data in the MS. As far as I can see from the methods, there is no compilation of data relating to ecosystem services provided by seagrasses – neither with regards of loss of ecosystem services or regarding potential re-establishment of services following recovery. I would therefore remove statements or rephrase (i.e. there's an expectation that with seagrass recovery there'll be a return of benefits, but that's not tested/ demonstrated here).
2. L229-L246: Consider using the PRISMA approach for reporting meta-analyses, including a flow diagram that details what records are kept in the analyses, which ones are excluded and why (www.prisma-statement.org)
3. L242: Vague statement. Please provide more details on 'data verification steps'
4. L280: add references/ values re: error of area/ depth limit assessment techniques?
5. Just a comment - Upon reaching the conclusion, I thought that an interesting follow up to this MS would be to undertake some modelling to estimate the timeline of future seagrass recovery – when may we expect to see seagrasses fully recover to mid 19th Century levels, based on known rates of recovery and further projected losses due to various factors ?
6. Although the MS is generally very well written, I recommend a final grammatical revision to polish the final text

Adriana Vergés

Reviewer #2 (Remarks to the Author):

The manuscript "Recent trend reversal for declining European seagrass meadows" is an encouraging paper for the entire field of seagrass ecology and marine conservation/restoration in general – showing that we can have some optimism that ecosystems can recover. It is an effective counterpoint to Waycott et al. 2009 (probably the most influential recently-published paper in seagrass ecology) so will most likely have a significant impact on academic studies and potentially on management practice with its optimistic message. I have a few suggestions for improvement, with one particularly critical suggestion and several minor ones.

MAJOR COMMENT

There is tension in the paper between two competing key points: (1) quantifying the various declines of European seagrass meadows over the past century and (2) that one extent metric – area – has increased in the past decade, although it appears to me that the authors want point (2) to be the key focus. This tension is felt most prominently in the Results and Discussion as follows. I could only find one sentence in the Results devoted to point (2) (Lines 139-141) which is buried towards the end of the second paragraph; whereas the entire rest of the Results is devoted to point (1). Conversely, the Discussion is the complete opposite – pretty much the whole Discussion is devoted to point (2). This means that several aspects of the Results associated with point (1) are not synthesised – for example, the authors state in Lines 130 and 131 that “16 sites accounted for 75% of the losses, whereas 5 sites accounted for the same percentage of gains”, for which a potential synthesis in the Discussion would be that gains are concentrated in a few key locations but losses are being seen across more sites (and in turn what does this mean for management), but since most of the statements in the Results are not explored in the Discussion, most of the Results are not synthesised in the paper. This in and of itself is not a problem, if in a revised version the Results and Discussion are better balanced in terms of how much emphasis they give to points (1) and (2).

However, more critically, this difference between focus of the Results and the Discussion means that there is not much analysis given to point (2). If point (2) is to be the key point of the paper, it should at the very least have its own subsection in the Results, and requires additional analysis or description/investigation to make clear to the reader (and the scientific community) that the trend reversal is real, substantial, and well-justified.

For example, this new subsection in the Results (and/or additional text in the Discussion) should address questions such as “why is area the only extent metric that increased?” e.g. why was this not seen in presence/absence – or was presence/absence metric converted to area? “If area is combined with density metrics (e.g. biomass/m²), has the total biomass of seagrass in Europe also increased in the last decade?” Or is such a calculation not possible due to the discrepancies in what is measured between the 200+ studies considered in the paper. Additionally, “were the area increases solely due to a few key sites having flourished, or many sites that increased in seagrass area by a small amount? Were there specific locations or countries or seas where the increase in seagrass area were concentrated?” This latter question lends itself to a map of changes – which the paper does have (Figure 1 and Figure S1) but only for the total seagrass changes over a century, not in the last decade, again illustrating the tension between points (1) and (2). Finally “for the changes in seagrass area observed in the last decade, what proportion were attributed to management vs natural recolonization?” Again this is addressed for point (1) but not I could not find discussion of it in point (2) – although the answer to the question appears to be shown in Figure 4B. Furthermore Lines 152-154 mention specific management actions that might have been responsible but these actions are not analysed in the paper as far as I can tell (hence their mention in Lines 152-154 should probably go to Discussion or be removed as they are not a Result of the analysis), although that would be an interesting thing to unpack as well – which management actions were considered to be the key ones which made a difference? (Although I recognise that answering this latter question might not be possible due to varying levels of synthesis amongst the 200+ studies – and if so the authors should state this.)

Alternatively, if the authors wish for both points (1) and (2) to be the key points of the paper, some re-balancing of the text in the Results and Discussion is still absolutely necessary - e.g. perhaps both sections could devote approximately equal amounts of text to points (1) and (2), or the Results are 20% point (1) 80% point (2) and the Discussion is 20% point (1) 80% point 2, etc. – depending on the authors' preference.

MINOR COMMENTS

1. The tension between points (1) and (2) is also felt in the title – the first half of the title (“Recent trend reversal”) is about point (2) but the second half of the title (“for declining European seagrass meadows”) is about point (1) – but this ends up being a bit tricky because it is a double-negative (reversal of decline). I also think “trend” should be replaced since the authors specifically saw only an increase in area. If point (2) is the key point, maybe “European seagrass meadow area increased over the last decade” (or something similar) might work? Overall I think the title is potentially ok as is, but a better title might be possible – I leave this up to the discretion of the authors.
2. Line 59 “gain and recovery” – choose one (“gains” or “recovery”)
3. Line 61 “uncertain” – implies large uncertainty bounds, perhaps change to “unknown” or “unclear”?
4. Line 66 change “trend-reversal” to “reversal”
5. Line 83 “in terms of the metrics of change assessed”
6. Line 85 change to “loss, gains or stability” or “loss, recovery or stability”
7. Line 91 and several other places throughout the manuscript – “extension” should be “extent”, so that this is not confused with leaf extension
8. Line 111 – state the extent and density metrics here so that the reader does not need to refer to the Methods
9. Line 140 and several other places throughout the manuscript – the reversal of decline in area is stated as “the first time in a century” but area changes are only shown (and calculated) from the 1950s onwards (Figure 3 and Line 305) – so this really should be the first time in 50 years. I recognise that this could potentially be misinterpreted as “seagrass increased in area in the 1940s” so some caveat statements around that this is the first area increase observed since sufficient data was available to quantify seagrass area trends at a continent scale might also be needed to avoid a different misinterpretation.
10. Lines 141-143 (see also major comment) – this sentence needs elaboration – does this mean that the observed seagrass recovery was primarily about (1) colonisation/recolonization of new/uninhabited areas suitable for seagrass, but (2) not much expansion of seagrass meadows to areas within the same site that were not suitable before and (3) not much increases in meadow health (e.g. biomass or cover)? This will help to clarify precisely what is the nature of the “reversal of decline” (which seems to be primarily about area). Also what about the trends in presence/absence? (this is not plotted on Figure 3) or was this metric somehow combined into area? I’d also potentially quibble here with “depth limits (upper and lower)” being classed as an extent metric – since it would only be an extent metric insofar as the environmental stressor of water quality and/or hydrodynamics - and it is potentially of benefit to the authors’ key point (2) to not class it as an extent metric – otherwise the reader might wonder why only 1 of the 3 extent metrics showed an increase in the 2000s. Having said that, I don’t necessarily think that depth limits need to be reclassified as a different type of metric, but perhaps some discussion around why depth limits are considered an extent metric in the Methods would be helpful – and perhaps text elsewhere clarifying that area is the best direct metric of extent (especially if it incorporates presence/absence data) so that key point (2) is given more strength.
11. Line 157 replace “raised as a” with “was the”
12. Line 166-167 since the species composition of the recent reversal was not discussed in the Results (see also major comment), the reader here does not know the evidence for the statement that this reversal was “mostly due to recovery of fast-growing species in some locations” – so text in the Results needs to be added with reference to Figure 4 so that it is clear what this statement refers to. Perhaps the text of Lines 210 to 211 should be moved to the Results, for example.
13. Line 181 change “elsewhere” to “outside”
14. Line 195 introduce the acronym WFD so that later references to WFD are clear
15. Lines 216-217 Authors could refer here to the seagrass species classifications introduced in Table 1 of Kilminster et al. 2015 *Sci. Total Environ.* 534: 97-109 (Zostera and Cymodocea are classified as opportunistic whilst Posidonia is classified as persistent; opportunistic species have faster turnover and

more rapid recovery rates than persistent species)

16. Lines 223-224 I think this final sentence could be made a bit more exciting. Is ecosystem services the only reason why we should restore seagrass? (If it is, what specific important ecosystem services provide a justification for restoring seagrass?)

17. Lines 240-242 It would be useful here to elaborate on what is the difference between the European datasets used in Waycott et al. (2009) vs this paper. Does this paper use all of the same datasets as Waycott et al. (2009) plus a lot more? (for example, does Waycott et al. 2009 only account for X% of the datasets included in the present study?)

18. Line 298 Would be useful to state here how much data (e.g. what X% of studies) was thrown out of the analysis because of the "at least 8 year time series data" rule.

19. Line 301 Clarify what is meant by "time-weighted mean"? Do you mean linear interpolation?

20. Lines 313-319 what distinguishes between "macroalgae cover due to eutrophication" and "macroalgae invasion" ? Is it just that macroalgae cover in the first case was a constant pressure?

21. Line 319 If possible can "sediment dynamics" be rephrased, because I am not sure how specifically this is an extreme event? Or is this just a secondary effect of the previously mentioned extreme events and if so maybe it should be deleted?

22. Figure 1 caption, Table 1 caption and Table S2 caption – include years across which the changes are being referred to (to avoid confusion with point (2))

23. Figure 2 – I found these difficult to interpret, maybe they should be replaced with histograms or bar charts?

24. Figure 3A – add dots to the data

25. Figure 3B – include a dashed line for 0 as was done in Figure 3A?

26. Figure 4B – I wasn't sure why certain time series (e.g. wasting disease) just stopped, I would think they would just go back down to zero? Or are these plots only showing dots for nonzero data? If so, the information for 1910s-1930s for mechanical damage might be misinterpreted as nonzero reports of mechanical damage because there is a line through this data but no points. So either remove the lines, or replace these figures by histograms?

27. Figure 4B why is there are blue dot for 2010s for natural recovery/colonisation but no information for green dot for 2010s for management intervention? I thought the analysis doesn't consider the 2010s. Or does the 2010 on the x-axis not equal the decade of the 2010s?

28. Figure S2 – I found this figure difficult to interpret, because of the log-scale, the vertical rescaling of "increase" and "decrease " sites to 100%, and because it is a figure within a figure. This figure might actually be unnecessary/redundant since Figure 2B already shows that the specific rate of decline exceeds the specific rate of increase.

Reviewer #3 (Remarks to the Author):

The manuscript represents a big effort on an important and interesting topic: trends in seagrass health and extent across Europe. However, much of the text is unclear and many terms are not defined. I'm not just talking about the English, although it certainly needs improving for both clarity and grammar, but the basis of the study. The authors are pulling together a vast amount of data from many sources and at many levels of detail, a difficult task, but they do not adequately indicate what parameters of change they are including in the analysis or the figures. Since they ultimately claim that recent efforts at coastal management have truly begun to reverse the seagrass losses of the past many decades, it is crucial that their documentation be accurate and believable. In its present form, the manuscript is too unclear to be published.

The manuscript is not improved over the previous version reviewed for a different journal. The data does not support the conclusions.

Edit for standard academic English – odd grammar to the point of being unclear, misuse of words
Misuse of “Remarkably” on line 176
Non-colloquial use of “reverted” on line 66ff
Sentence running from lines 212 – 214 unclear, as are many others

Line 217, 224, 226, many other places AND in the figures - *Zostera noltii* spelled incorrectly as *Zostera noltei*

Unclear?????? Are they reporting a slow down of loss or an overall gain????

Figure 1 – must give a time frame and we must be told whether the wasting disease decline and recovery is represented in Figure 1. This figure does not seem to match the discussion in the text of where and how much seagrass has recovered across Europe.

Figure 3 – should read “rate of decline” and “rate of increase”

Most of the gain in seagrass area is due either to *Z. noltii* spreading in the Wadden Sea or to the recovery of *Z. marina* after the wasting disease epidemic – so how can you attribute the gains to better management?

Mention of global warming as probably impacting any future gains of seagrass is weak

Define “meadow” – is it a unit that is reported on at a certain time, is it a given size, is it an embayment or other geographical limitation? How many seagrass meadows are there in Europe?

Line 105, 110,... Authors imply that there has been a 147 year record for all or many seagrass meadows, whereas the long record may be only for Denmark.

Including or excluding the wasting disease data has a complicated effect on the change and rates of change. For example, in Fig 3A, the wasting disease is not included in the “declining rate” because the 1930s are not included, making the decline rate only ~ -8, whereas inclusion of the 1930s and the wasting disease in Figure 3C inflates the number of lost meadows. The wasting disease of the 1930s should be consistently included or excluded from the study.

“RESPONSE TO REFEREES” LETTER FOR SUBMISSION NCOMMS-18-36760.

Black text – Point requiring a response

Blue text – Author response

REPLY TO REVIEWER #1

Reviewer #1 (Remarks to the Author):

In this MS, de los Santos and colleagues review changes in seagrass cover in European waters over the last century and a half. The data shows that over the last 147 years there's been a loss of about a third of seagrass cover. The authors identify a period of peak loss in the 1970s- 1980s, as well as a reversal in the overall trend of decline in the 2000s, when seagrass recovery rates become faster than seagrass loss. All of this is complemented with data on the causes leading to both declines and recovery.

This is an exciting and timely MS that should be of great relevance to conservation biologists broadly. The exceptional long temporal span and broad spatial scale provides a unique and highly valuable perspective that invites some optimism with regards to the future state of seagrass meadows, as the data demonstrates that many of the recent management interventions to stop seagrass decline are working effectively. This paper is likely to become highly influential among both conservationists and marine ecologists.

I have minor comments/ recommendations that I think would strengthen the MS, which I detail below:

1. L69 Abstract and L223-L224 – Statement that recovery of seagrass is bringing back services and benefits provided by seagrasses is not supported by the data in the MS. As far as I can see from the methods, there is no compilation of data relating to ecosystem services provided by seagrasses – neither with regards of loss of ecosystem services or regarding potential re-establishment of services following recovery. I would therefore remove statements or rephrase (i.e. there's an expectation that with seagrass recovery there'll be a return of benefits, but that's not tested/ demonstrated here).

ACTION: We have rephrased both the Abstract (line 71) and Discussion sentences (lines 255-257) according to the reviewer's suggestion.

2. L229-L246: Consider using the PRISMA approach for reporting meta-analyses, including a flow diagram that details what records are kept in the analyses, which ones are excluded and why (www.prisma-statement.org).

REPLY: We are aware that using the PRISMA approach is convenient to report the records tracked in a meta-analysis. Unfortunately, we are unable to state the number of sources identified or screened in the earliest stages of the study, since that information was not saved at the time (we could repeat the searches in Web of Science, but this would be pretty time-consuming at this stage).

ACTION: We have improved the reproducibility and clarity of the meta-analysis by stating (lines 270-274) the number of sources assessed for eligibility (n = 520) (3rd step in PRISMA statement), the number of sources excluded (n = 279) and their reasons, and the number of sources included in the final compilation (n = 241). The reasons for exclusion were: 1) inaccessible source, 2) source including a site already compiled or updated in another more recent source, 3) source being a review or compilation (in those cases, the source was consulted to find potential studies to assess); and, 4) sources in which data criteria were not met (e.g. type of metrics).

3. L242: Vague statement. Please provide more details on 'data verification steps'

ACTION: We have specified the verification steps conducted (lines 281-283).

4. L280: add references/ values re: error of area/ depth limit assessment techniques?

ACTION: The reference of Waycott et al. (2009) – i.e. referece [4] – was added (line 320).

5. Just a comment - Upon reaching the conclusion, I thought that an interesting follow up to this MS would be to undertake some modelling to estimate the timeline of future seagrass recovery – when may we expect to see seagrasses fully recover to mid 19th Century levels, based on known rates of recovery and further projected losses due to various factors?

REPLY: We do appreciate this interesting suggestion and will consider a follow-up focus on the recovery time in future works.

ACTION: None.

6. Although the MS is generally very well written, I recommend a final grammatical revision to polish the final text

ACTION: A final grammatical revision has been made throughout the entire MS.

Adriana Vergés

Reviewer #2 (Remarks to the Author):

The manuscript “Recent trend reversal for declining European seagrass meadows” is an encouraging paper for the entire field of seagrass ecology and marine conservation/restoration in general – showing that we can have some optimism that ecosystems can recover. It is an effective counterpoint to Waycott et al. 2009 (probably the most influential recently-published paper in seagrass ecology) so will most likely have a significant impact on academic studies and potentially on management practice with its optimistic message. I have a few suggestions for improvement, with one particularly critical suggestion and several minor ones.

MAJOR COMMENT

There is tension in the paper between two competing key points: (1) quantifying the various declines of European seagrass meadows over the past century and (2) that one extent metric – area – has increased in the past decade, although it appears to me that the authors want point (2) to be the key focus. This tension is felt most prominently in the Results and Discussion as follows. I could only find one sentence in the Results devoted to point (2) (Lines 139-141) which is buried towards the end of the second paragraph; whereas the entire rest of the Results is devoted to point (1).

Conversely, the Discussion is the complete opposite – pretty much the whole Discussion is devoted to point (2). This means that several aspects of the Results associated with point (1) are not synthesised – for example, the authors state in Lines 130 and 131 that “16 sites accounted for 75% of the losses, whereas 5 sites accounted for the same percentage of gains”, for which a potential synthesis in the Discussion would be that gains are concentrated in a few key locations but losses are being seen across more sites (and in turn what does this mean for management), but since most of the statements in the Results are not explored in the Discussion, most of the Results are not synthesised in the paper. This in and of itself is not a problem, if in a revised version the Results and Discussion are better balanced in terms of how much emphasis they give to points (1) and (2).

REPLY: This is a very good suggestion that was well considered. In fact, we think that both points are important and have balanced them in the revised MS.

ACTION: We increased the proportion of point 2 in the Results (lines 141-160) and of point 1 in the Discussion (lines 187-201). See next comment for details of the text added in each section.

However, more critically, this difference between focus of the Results and the Discussion means that there is not much analysis given to point (2). If point (2) is to be the key point of the paper, it should at the very least have its own subsection in the Results and requires additional analysis or description/investigation to make clear to the reader (and the scientific community) that the trend reversal is real, substantial, and well-justified.

REPLY: We agree that additional analyses is beneficial to support and understand the reversal in the declining trajectory of seagrasses in Europe during the 2000s.

ACTION: With this aim, we included two new analyses in the MS:

- 1) The assessment of the number of reports of sites where the trajectory improved, worsened or stayed steady from decade to decade, that is the trajectory evolution, following the criteria indicated in the table:

First decade	Second decade	TRAJECTORY EVOLUTION
No change	No change	Steady
No change	Declining	Worsening
No change	Increasing	Improving
Declining	No change	Improving
Declining	Declining	Worsening
Declining	Increasing	Improving
Increasing	No change	Steady
Increasing	Declining	Worsening
Increasing	Increasing	Improving

The analysis was conducted both area, density metrics (cover, shoot density and biomass) and depth limits (lower and upper), as suggested in the following reviewer's comment. A new Figure (Figure 4B) was produced with the decadal frequency of trajectories (improving, worsening, steady), which highlights further the trend reversal that occurred in the 2000s. The number of sites experiencing an area trajectory improvement increased and surpassed the number of those worsening, supporting the 2000s trend reversal of the decadal rate of change in area previously shown in Figure 3. Concerning the trajectory of density metrics, the pattern found during the 1990s and the 2000s was the prevalence of trajectories improving or in steady state, even though in the 2000s there was an increase of worsening trajectories.

- 2) The species-specific decadal trajectory of the area rate of change (new Figure S3 in SI), as previously done for the whole set of species (Figure 3). This analysis revealed that the trend reversal detected in the 2000s was due to the slowdown of losses of all the species except *C. nodosa*, which losses surpassed gains in the 1990s and 2000s, along with the fast area recovery of *Z. marina* and *Z. noltei* in the 2000s.

The description of the new analyses was included in Materials and Methods (lines 337-346 and line 347), the results were added (lines 151-154 and lines 156-158) with new figures (Figure 4B and Figure S3 in the SI) and new text was included in the Discussion (lines 205-210). Panels of Figure 4A correspond to the old Figure 3B but using box-plots and adding the area, so the three groups of

metrics can be compared easier in terms of specific rate of change and evolution of trajectory. The change in data visualization was done to follow the points of the Nature Communications Editorial Policy Checklist.

For example, this new subsection in the Results (and/or additional text in the Discussion) should address questions such as “why is area the only extent metric that increased?” e.g. why was this not seen in presence/absence – or was presence/absence metric converted to area? “If area is combined with density metrics (e.g. biomass/m²), has the total biomass of seagrass in Europe also increased in the last decade?” Or is such a calculation not possible due to the discrepancies in what is measured between the 200+ studies considered in the paper.

REPLY: The combination and comparison of metrics is not straightforward because extent and density metrics are rarely jointly assessed in the same sites (only 9 % of the sites compiled included both type of metrics). For this reason, we decided not to inter-convert or combine metrics as the reviewer suggested in order to avoid errors and uncertainty in the results. However, with the new analysis explained above, we can compare the evolution over the last decades in area, depth limits and density metrics, in a qualitative way.

ACTION: We have discussed the differences in the trends for area and density in the Discussion (lines 187-192).

Additionally, “were the area increases solely due to a few key sites having flourished, or many sites that increased in seagrass area by a small amount? Were there specific locations or countries or seas where the increase in seagrass area were concentrated?” This latter question lends itself to a map of changes – which the paper does have (Figure 1 and Figure S1) but only for the total seagrass changes over a century, not in the last decade, again illustrating the tension between points (1) and (2).

REPLY: We think that there are already many figures both in the main text and as SI and decided to address this subject along the text.

ACTION: The nature of the gains in area during the 2000s are now explained in terms of species composition and regions (lines 146-153).

Finally, “for the changes in seagrass area observed in the last decade, what proportion were attributed to management vs natural recolonization?” Again, this is addressed for point (1) but not I could not find discussion of it in point (2) – although the answer to the question appears to be shown in Figure 4B. Furthermore Lines 152-154 mention specific management actions that might have been responsible but these actions are not analysed in the paper as far as I can tell (hence

their mention in Lines 152-154 should probably go to Discussion or be removed as they are not a Result of the analysis), although that would be an interesting thing to unpack as well – which management actions were considered to be the key ones which made a difference? (Although I recognise that answering this latter question might not be possible due to varying levels of synthesis amongst the 200+ studies – and, if so, the authors should state this.)

REPLY: It is impossible to assess the reviewer’s question “for the changes in seagrass area observed in the last decade, what proportion were attributed to management vs natural recolonization?” because most sources do not report the causes of change. We recognise that there is a considerable level of uncertainty in the original sources that is transferred into our analysis, especially when trying to associate the causes of change identified for each decade (Figure 5B) with the changes in area over decades (Figure 3 or Figure 4), since the data shown in the figures did not necessarily come from the same sites. This is the reason why we cannot explore further which management actions were the key ones to make a difference.

The management actions referred in lines 152-154 of the previous version (“improvement of water quality, re-established salinity regimes, and regulating anchoring and trawling”) are part of the causes analysed in the paper, as explained in the categories used in materials and methods (lines 367-370). The “established salinity regime” is a particular case of seagrass gains in which a rapid colonisation occurred after seagrass have been vanished due to two consecutive floods that decreased the salinity and increased the water turbidity in a coastal lagoon. This was incorrectly placed in the sentence since it was a case of natural recovery.

ACTION: The fact that most sources do not report the causes of change and thus that it is difficult to associate specific management actions to seagrass recovery has been clarified in the Discussion (lines 210-211 and lines 231-232). Cases when this association was possible were referred in the text (lines 211-219). The “reestablishment of salinity regime” has been moved to the line where natural recovery is explained (line 172).

Alternatively, if the authors wish for both points (1) and (2) to be the key points of the paper, some re-balancing of the text in the Results and Discussion is still absolutely necessary - e.g. perhaps both sections could devote approximately equal amounts of text to points (1) and (2), or the Results are 20% point (1) 80% point (2) and the Discussion is 20% point (1) 80% point 2, etc. – depending on the authors’ preference.

ACTION: As explained before, we have rewritten the results and discussion sections, so they now present a more even proportion of text dedicated to each point.

MINOR COMMENTS

1. The tension between points (1) and (2) is also felt in the title – the first half of the title (“Recent trend reversal”) is about point (2) but the second half of the title (“for declining European seagrass meadows”) is about point (1) – but this ends up being a bit tricky because it is a double-negative (reversal of decline). I also think “trend” should be replaced since the authors specifically saw only an increase in area. If point (2) is the key point, maybe “European seagrass meadow area increased over the last decade” (or something similar) might work? Overall, I think the title is potentially ok as is, but a better title might be possible – I leave this up to the discretion of the authors.

REPLY: We appreciate the reviewer’s suggestions, but we think that the title is catchy and summarises adequately the two findings of our study: the prevalence of declines in the past and the recent reversal.

ACTION: None.

2. Line 59 “gain and recovery” – choose one (“gains” or “recovery”)

ACTION: Done (line 59).

3. Line 61 “uncertain” – implies large uncertainty bounds, perhaps change to “unknown” or “unclear”?

ACTION: Changed to “unclear” (lines 61 and 93).

4. Line 66 change “trend-reversal” to “reversal”

ACTION: Done (line 66).

5. Line 83 “in terms of the metrics of change assessed”

ACTION: Done (line 85).

6. Line 85 change to “loss, gains or stability” or “loss, recovery or stability”

ACTION: Done (line 87).

7. Line 91 and several other places throughout the manuscript – “extension” should be “extent”, so that this is not confused with leaf extension

ACTION: Done (e.g. lines 90, 93).

8. Line 111 – state the extent and density metrics here so that the reader does not need to refer to the Methods

ACTION: Done (lines 112-113).

9. Line 140 and several other places throughout the manuscript – the reversal of decline in area is stated as “the first time in a century” but area changes are only shown (and calculated) from the 1950s onwards (Figure 3 and Line 305) – so this really should be the first time in 50 years. I recognise that this could potentially be misinterpreted as “seagrass increased in area in the 1940s” so some caveat statements around that this is the first area increase observed since sufficient data was available to quantify seagrass area trends at a continent scale might also be needed to avoid a different misinterpretation.

REPLY: We agree with the reviewer that the decadal trends refer to the last 60 years (1950s to 2000s) and not the whole century, due to the lack of enough data before 1950s to assess the trends.

ACTION: The time framework for the area calculations and other decadal analyses has been corrected throughout the whole MS (e.g. lines 142, 183), and legends of tables and figures. In addition, a caveat statement on data deficiency before the 1950s has been included (lines 204-205) to ensure that misinterpretations are avoided.

10. Lines 141-143 (see also major comment) – this sentence needs elaboration – does this mean that the observed seagrass recovery was primarily about (1) colonisation/recolonization of new/uninhabited areas suitable for seagrass, but (2) not much expansion of seagrass meadows to areas within the same site that were not suitable before and (3) not much increases in meadow health (e.g. biomass or cover)? This will help to clarify precisely what is the nature of the “reversal of decline” (which seems to be primarily about area).

REPLY: Based on the new analysis of evolution of trajectories over decades, we can now say that density metrics (cover, shoot density and cover), and not only area, showed an improvement in the 2000s. Old Figure 3B (for which the comment was related to) showed the average of the rate of change per decade, but without considering the separation by trajectory (decline, increase or no change), because the sample size per decade was not sufficient to do the analysis per trajectory. For clarity, we redid this figure (Figure 4A) using boxplots and showing the specific rate of change for area, density metrics and depth limits.

ACTION: None.

Also, what about the trends in presence/absence? (this is not plotted on Figure 3) or was this metric somehow combined into area?

REPLY: We have considered the assessment of the trend in presence/absence, by counting the number of sites per decade in which seagrasses disappeared or in which the seagrasses re-appeared. However, we found this assessment to be tricky because some time-series cover more than a decade with just two observations, making it difficult to determine when the reappearance or total disappearance exactly occurred. For instance, Bull et al. 2010 conducted a survey in Gibraltar in 2008, stating that all seagrasses had disappeared, but the only previous study reporting seagrasses in the same area had been done in 1993. Thus, we cannot infer in which decade (1990s or 2000s) the seagrasses were lost.

We did not combine this metric with area, so the only assessment in which presence/absence was included was in the overall trajectory (Figure 1 and Figure S1).

ACTION: None.

I'd also potentially quibble here with "depth limits (upper and lower)" being classed as an extent metric – since it would only be an extent metric insofar as the environmental stressor of water quality and/or hydrodynamics - and it is potentially of benefit to the authors' key point (2) to not class it as an extent metric – otherwise the reader might wonder why only 1 of the 3 extent metrics showed an increase in the 2000s. Having said that, I don't necessarily think that depth limits need to be reclassified as a different type of metric, but perhaps some discussion around why depth limits are considered an extent metric in the Methods would be helpful – and perhaps text elsewhere clarifying that area is the best direct metric of extent (especially if it incorporates presence/absence data) so that key point (2) is given more strength.

REPLY: Depth limits were classified as extent metric because when they change, seagrass area will change (for example, a regression of the lower depth limit from -18 m to -15 m will mean a decrease in seagrass areal extent that will depend on the slope of the meadow). We are aware that different metrics may be associated to specific environmental stressors, but it is far beyond our objectives picking metrics depending on the causes of change to assess. Indeed, the selection of metrics is particularly difficult because the use of metrics seems to be related with the methodologies available at different European regions (e.g. depth limits are widely used in the Baltic Sea).

As explained in another reply, conversion of presence/absence to area was not done because it would result in high uncertainties. Finally, we used area as the most important metric for the decadal analysis because it is the metric with more time-series available.

ACTION: None.

11. Line 157 replace “raised as a” with “was the”

ACTION: Done (line 173).

12. Line 166-167 since the species composition of the recent reversal was not discussed in the Results (see also major comment), the reader here does not know the evidence for the statement that this reversal was “mostly due to recovery of fast-growing species in some locations” – so text in the Results needs to be added with reference to Figure 4 so that it is clear what this statement refers to. Perhaps the text of Lines 210 to 211 should be moved to the Results, for example.

ACTION: We included in the results the contribution of the different species to the changes in seagrass area during the 2000s (lines 146-153) as well as in Figure S3.

13. Line 181 change “elsewhere” to “outside”

ACTION: Done (line 228).

14. Line 195 introduce the acronym WFD so that later references to WFD are clear

ACTION: Done (line 235).

15. Lines 216-217 Authors could refer here to the seagrass species classifications introduced in Table 1 of Kilminster et al. 2015 Sci. Total Environ. 534:97-109 (Zostera and Cymodocea are classified as opportunistic whilst Posidonia is classified as persistent; opportunistic species have faster turnover and more rapid recovery rates than persistent species).

REPLY: We appreciate this suggestion. However, a reference was already included regarding the same type of species classification (reference 46: O'Brien KR et al. (2017). Seagrass ecosystem trajectory depends on the relative timescales of resistance, recovery and disturbance. Mar Pollut Bull 134:166–176.).

ACTION: None.

16. Lines 223-224 I think this final sentence could be made a bit more exciting. Is ecosystem services the only reason why we should restore seagrass? (If it is, what specific important ecosystem services provide a justification for restoring seagrass?)

RESPONSE: We agree that a better statement could be done on the recovery of function and services after seagrass restoration. However, reviewer #1 raised a criticism on this point because we did not quantify the recovery of seagrass ecosystem services in our study. Based on her comment, we would like to be more cautious in this sentence, so we rephrased it accordingly.

ACTION: We rephrased the sentence (lines 256-257).

17. Lines 240-242 It would be useful here to elaborate on what is the difference between the European datasets used in Waycott et al. (2009) vs this paper. Does this paper use all of the same datasets as Waycott et al. (2009) plus a lot more? (for example, does Waycott et al. 2009 only account for X% of the datasets included in the present study?)

REPLY: Waycott et al. 2009 included 34 sites from Europe (out of 483 sites in their compilation) with time-series on seagrass area extent, being 3 of them duplicated: Vaccares lagoon (site reference number in Waycott's dataset 38 and 78), Swedish west coast (site reference 26-29 and 335), and Glenan Archipelago (site reference number 88 and 339). Thus, we included the 31 unique sites from Waycott's dataset, for which updates on area extent were only available in 13 of them. In regard to the specific question made by the reviewer, we used all the European sites in Waycott's dataset (31) plus 706 sites more, that is, Waycott's dataset only account for 4% of the sites included in the present study.

ACTION: Details on the number of sites in regard to Waycott's dataset have been included in lines 278-280.

18. Line 298 Would be useful to state here how much data (e.g. what X% of studies) was thrown out of the analysis because of the "at least 8-year time series data" rule.

REPLY: We agree. We compiled 315 time-series (43 % of the all compiled sites) that met the 8-yr criterium.

ACTION: We have included this information in line 337. In addition, we have included another resource in the supplementary information (Supplementary Information 3) showing the time-series for the compiled sites, which helps visualize those used in the decadal analysis.

19. Line 301 Clarify what is meant by "time-weighted mean"? Do you mean linear interpolation?

REPLY: When doing the decadal analysis, some decades (first or last one) may be incomplete (see new resource in the supplementary material S3). For instance, shoot density may have changed from 1957-1960 (1950s), 1960-1970 (1960s), 1970-1978 (1970s) for a site. This implies 3 decades with a rate of change (μ) associated to each of them, even though the time coverage within each decade is not the same (time = 3, 10 and 8 years, respectively). When putting together all the rates of change for a specific decade, we used the time-weighted mean to calculate the average rate of change for that specific decade: e.g. rate of change for 1950s = $[\mu_1 * t_1 + \mu_2 * t_2 + \dots + \mu_n * t_n] / \text{sum}(t_1 + t_2 + \dots + t_3)$. In this way, a rate of change calculated for the whole decade (10 years) will have a greater contributor to the average than a rate of change calculated for a fraction of that decade.

ACTION: Since we redid the plots for density and depth limits metrics as boxplots, the weighted average was not used anymore so this explanations was deleted from the MS.

20. Lines 313-319 what distinguishes between “macroalgae cover due to eutrophication” and “macroalgae invasion”? Is it just that macroalgae cover in the first case was a constant pressure?

REPLY: “Macroalgae cover” due to eutrophication refers to the overgrowth of algae (e.g. filamentous algae) due to high inputs of nutrients in the system (e.g. Langstone Harbour, UK, den Hartog et al. 1994), while “macroalgae invasion” refers to the growth of introduced macroalgae species, for example growth of *Lophocladia lallemandii* on *Posidonia oceanica* (Ballesteros et al. 2007).

ACTION: We have clarified the differences between the two causes by referring to them as “macroalgae overgrowth due to eutrophication” (lines 359-360) and “non-native macroalgae effects” (line 363).

21. Line 319 If possible, can “sediment dynamics” be rephrased, because I am not sure how specifically this is an extreme event? Or is this just a secondary effect of the previously mentioned extreme events and if so, maybe it should be deleted?

ACTION: The cause of change “sediment dynamics” has been deleted and the primary cause has been identified and re-categorised accordingly. Table S2 and S3 have been updated following this change.

22. Figure 1 caption, Table 1 caption and Table S2 caption – include years across which the changes are being referred to (to avoid confusion with point (2)).

ACTION: We have clarified in those captions the time windows they refer to.

23. Figure 2 – I found these difficult to interpret, maybe they should be replaced with histograms or bar charts?

ACTION: Figure 2 was redone using stacked bars (for proportion of declining and decreasing sites) and boxplots (for specific rates of change).

24. Figure 3A – add dots to the data.

ACTION: Done.

25. Figure 3B – include a dashed line for 0 as was done in Figure 3A?

ACTION: Done (now Figure 4A).

26. Figure 4B – I wasn't sure why certain time series (e.g. wasting disease) just stopped, I would think they would just go back down to zero? Or are these plots only showing dots for nonzero data? If so, the information for 1910s-1930s for mechanical damage might be misinterpreted as nonzero reports of mechanical damage because there is a line through this data but no points. So, either remove the lines, or replace these figures by histograms?

REPLY: The plots only show non-zero data that is why some time-series seems to stop. We agree that the line should only connect the dots of consecutive decades with non-zero data.

In addition, we noticed that we should only focus on the 1950s-2000s time coverage to be consistent with the other decadal analysis.

ACTION: The lines connecting non-consecutive decades were removed. The figure (now figure 5B) was limited to the 1950s-2000s time range. Information that the plots only show non-zero data and that lines connect dots of consecutive decades with non-zero data was included in the figure legend.

27. Figure 4B why is there are blue dot for 2010s for natural recovery/colonisation but no information for green dot for 2010s for management intervention? I thought the analysis doesn't consider the 2010s. Or does the 2010 on the x-axis not equal the decade of the 2010s?

REPLY: As explained in the previous reply, we decided to exclude the decade 2010s for being incomplete and to be consistent with the other decadal analysis.

ACTION: The figure (now figure 5B) was altered accordingly.

28. Figure S2 – I found this figure difficult to interpret, because of the log-scale, the vertical rescaling of “increase” and “decrease” sites to 100%, and because it is a figure within a figure. This figure might actually be unnecessary/redundant since Figure 2B already shows that the specific rate of decline exceeds the specific rate of increase.

REPLY: We would like to keep this figure because it shows a different information to that shown in Figure 2B: while Figure S2 shows the frequency of area losses and increases depending on their “size”, i.e. few big losses/gains and many small losses/gains), Figure 2B shows how fast the changes in area (and other metrics) occurred.

ACTION: None.

Reviewer #3 (Remarks to the Author):

The manuscript represents a big effort on an important and interesting topic: trends in seagrass health and extent across Europe. However, much of the text is unclear and many terms are not defined. I'm not just talking about the English, although it certainly needs improving for both clarity and grammar, but the basis of the study. The authors are pulling together a vast amount of data from many sources and at many levels of detail, a difficult task, but they do not adequately indicate what parameters of change they are including in the analysis or the figures. Since they ultimately claim that recent efforts at coastal management have truly begun to reverse the seagrass losses of the past many decades, it is crucial that their documentation be accurate and believable. In its present form, the manuscript is too unclear to be published.

The manuscript is not improved over the previous version reviewed for a different journal. The data does not support the conclusions.

REPLY: We really do not understand this comment as the manuscript was indeed thoroughly amended based on the useful reviewers' comments from the submission (and rejection) to the previous journal. Furthermore, some of the comments below refer to the previous version that the reviewer read and not to the one submitted to Nature Communications. Perhaps the reviewer confounded both submissions.

In any case, the present revised manuscript does not claim anymore that coastal management efforts were responsible for seagrass recovery. This is only discussed as an explaining hypothesis. In addition, we added an extra analysis (see reply to reviewer #2) showing that two metrics (area and density) show a reversal trend in the 2000s and that this is not only due to the spreading of *Z. noltei* in the Wadden Sea as commented below by the reviewer. We are willing to provide more supportive analysis if detailed criticisms on the speculated mismatching of data analysis and conclusions are provided by the reviewer, since his/her statement is pretty general.

ACTION: Following the comments of reviewer #2, we added a new data analysis (Figure 4B and Figure S3 in the SI) and revised the text of Materials and Methods (lines 342-347), Results (lines 146-154 and lines 157-158) and Discussion (lines 205-208) accordingly.

Edit for standard academic English – odd grammar to the point of being unclear, misuse of words.

ACTION: A careful grammatical revision has been made.

Misuse of “Remarkably” on line 176.

REPLY: This word was not present in the manuscript submitted to Nature Communications.

ACTION: None.

Non-colloquial use of “reverted” on line 66ff

ACTION: The verb “to revert” in lines 224 has been replaced by “to reverse”.

Sentence running from lines 212 – 214 unclear, as are many others

REPLY: The original lines 212-214 in the first submitted version of the MS to Nature communication were: “Species-specific recovery times are expected, from a few years for *Z. noltei* and *C. nodosa* to a decade for *Z. marina* and to a century for *P. oceanica*, depending on their rhizome extension rates, branching rates and branching angles (46), and on the rate of formation of new patches (47).” However, this sentence was deleted from discussion in order to get sound length for this section after addition on new text following reviewer #2’s comments.

ACTION: None.

Line 217, 224, 226, many other places AND in the figures - *Zostera noltii* spelled incorrectly as *Zostera noltei*.

REPLY: *Zostera noltei* is the currently accepted spelling for this species, as stated in WoRMS (<http://www.marinespecies.org/aphia.php?p=taxdetails&id=145796>) and ALGAEBASE (http://www.algaebase.org/search/species/detail/?species_id=f2f6f3fd623b02f81).

ACTION: None.

Unclear?????? Are they reporting a slowdown of loss or an overall gain????

REPLY: We are not completely sure to which specific part of the MS the reviewer is referring to as “unclear”, but it is probably related to Figure 3A. In this figure, we show the decadal rate of change in area (% decade⁻¹) separately for the sites categorised as “declining” and “increasing”, and then the net specific rate of change when sites in both categories are computed together. The results indicate a change in the losses (red line) from the peak in the 1970s (-33.6 % dec⁻¹) to less negative rates in the 1980s (-27.0 % dec⁻¹), in the 1990s (-16.1 % dec⁻¹) and in the 2000s (-8.3 % dec⁻¹), which can be regarded as a slowdown because the rates become less negative along the decades. In other words, the specific rate of change improved +6.6% dec⁻¹ from 1970s to 1980s, + 11.0 % dec⁻¹ from 1980s to 1990s, and +7.8 % dec⁻¹, from 1990s to 2000s. This slowdown is not only a consequence of the gains, since the two type of trajectories are assessed separately.

ACTION: We included an explanation of why we considered that the decadal rate of change in area slowed down from 1980s onwards (lines 141-146).

Figure 1 – must give a time frame and we must be told whether the wasting disease decline and recovery is represented in Figure 1. This figure does not seem to match the discussion in the text of where and how much seagrass has recovered across Europe.

REPLY: The map in Figure 1 shows the compilation of seagrass trajectories for the 737 sites compiled, independently of the time in which their metrics were assessed, and using the overall analysis approach, that is, using the initial and final observation over the time window covered in each study (as explained in Materials and Methods, see lines 314-315). Those trajectories are based on the available time information for each site at different time windows between 1869 to 2016 so do not necessarily reflect present or synchronic trends.

The declines due to wasting disease in the 1930s are depicted (e.g. most of the red circles along the Atlantic French coast, and others in the Netherlands and UK), and so are the sites experiencing recoveries. We would like to note that some sites that experienced recovery but have not yet returned to the initial state, are plotted as decline in Figure 1. The discussion of where and how much seagrass has recovered across Europe is not based on Figure 1, but on the decadal analysis (Figure 3 and Figure 4).

ACTION: We have clarified in Figure 1 caption the time window of the map.

Figure 3 – should read “rate of decline” and “rate of increase”

ACTION: Figure was altered following the reviewer’s suggestion.

Most of the gain in seagrass area is due either to *Z. noltii* spreading in the Wadden Sea or to the recovery of *Z. marina* after the wasting disease epidemic – so how can you attribute the gains to better management?

REPLY: It is true that great part of the gains in seagrass area is due to the spreading of *Z. noltei* in the Wadden Sea, but there are other sites and species (mostly *Z. marina*) that contributed to these gains that may be related to management efforts (see reply to reviewer #2’s comment).

In relation to the reasons behind the spreading of *Z. noltei* in the Wadden Sea, Dolch et al. 2017 considered the following: “Intertidal beds of narrow-leaved *Z. marina* and *Z. noltii* declined during the 1970-80s presumably caused by anthropogenic eutrophication (de Jonge & de Jong, 1992). Especially the southwestern and central Wadden Sea were affected as these regions are in proximity to the big estuaries (van Katwijk et al., 2000; Dolch et al., 2013; Folmer et al., 2016).

Riverine nutrient inputs to the Wadden Sea have been reduced for the last 25-30 years which has particularly benefited seagrass bed recovery in the northern Wadden Sea since the mid-1990s.”

ACTION: We have clarified the causes of the spreading of *Z. noltei* in the Northern Wadden Sea (lines 211-213).

Mention of global warming as probably impacting any future gains of seagrass is weak

REPLY: This comment was in the first revision that the reviewer did, and we had already removed that sentence before the submission to Nature Communications.

ACTION: None.

Define “meadow” – is it a unit that is reported on at a certain time, is it a given size, is it an embayment or other geographical limitation? How many seagrass meadows are there in Europe?

REPLY: “Meadow” is not the unit of study, it is “seagrass site”, which is an area where seagrasses occur, which may vary in size, and which geographical limitation was defined in the source (sometimes it is a monitoring point in a wider seagrass area, sometimes, it is an isolated continuous meadow, sometimes it is a confined area such as a bay or lagoon, etc.). We cannot answer the reviewer’s question “How many seagrass meadows are there in Europe?” because it is out of the scope of our study, in which we did not compiled distributional data but studies showing temporal trends of seagrasses in Europe.

ACTION: We have checked the MS to correct any misuse of “meadow” as unit of study.

Line 105, 110, ... Authors imply that there has been a 147-year record for all or many seagrass meadows, whereas the long record may be only for Denmark.

REPLY: The lines referred by the reviewer are not from the version submitted to Nature Communications.

ACTION: This subject was further clarified using the following statement (lines 293-295) “The overall dataset covered 147 years, from 1869 to 2016, with the observation effort increasing exponentially over time. The time series was highly variable among sites, from 1 to 121 years with a median of 9 years”.

Including or excluding the wasting disease data has a complicated effect on the change and rates of change. For example, in Fig 3A, the wasting disease is not included in the “declining rate” because the 1930s are not included, making the decline rate only ~ -8, whereas inclusion of the 1930s and

the wasting disease in Figure 3C inflates the number of lost meadows. The wasting disease of the 1930s should be consistently included or excluded from the study.

REPLY: Figure 3C does not exist in the MS version submitted to Nature Communications. When amending the previous version, we realised that it was complicated to compare trends and descriptors of change before the 1950s as they were inflated with qualitative data such as presence/absence. Consequently, the version submitted to Nature Communications focus on the decadal changes from the 1950s onwards.

ACTION: None.

ADDITIONAL CHANGES

In addition to the changes done following the reviewers' comments and suggestions, we have done some clarifications and corrections during the proof-reading of the amended version:

- Legends of figures and tables were checked for clarity.
- Names of variables (e.g. decadal rate of change of area) and trajectories were checked for consistency in the figures and text.
- We have deleted the paragraph in the discussion of the previous version about seagrass recovery (old lines 204-217) to reduce the number of words of this section after the inclusion of new text following reviewer #2's comments.
- Citations and reference list were updated.
- Figure 5. The area losses were deleted because we realised that it was not mentioned in the MS and it was already included in Table S2.
- Figure S1. We have corrected this table because it was from a previous submission to another journal and it had not been updated after the changes done before submission to Nature Communications. Now, it matches the rates given in Figure 3 and in the MS. The table has been also reduced now so it only includes data from the 1950s to the 2000s, for consistency with the decadal analysis presented.
- We have carefully revised the MS in response to the points on the editorial policy checklist and reporting summary (sample size, type of plots and statistical reports).

REVIEWERS' COMMENTS:

Reviewer #2 (Remarks to the Author):

I applaud the authors of the manuscript "Recent trend reversal for declining European seagrass meadows" for their careful and thorough consideration of all reviewers' comments. The manuscript has significantly improved. I have some additional (minor) suggestions to improve the manuscript; mostly related to the changes they have made, after which I believe that the paper is suitable for publication in Nature Communications.

1. Abstract Line 68 If this statement is related to Figure 4B, it should be "the rate of change of seagrass density"
2. Lines 126-139 I still find Figure S2 very unclear. I don't believe it helps to justify, or assist in the understanding of, the results mentioned in the text. I cannot understand how Figure S2 shows that 16 sites accounted for 75% of the losses and 5% sites accounted for the same percentage of gains (Lines 128-129). I also cannot understand how Figure S2 demonstrates a skewed distribution of the frequencies of losses and gains because the log scale on the x-axis hides the skew. Please consider another figure to justify these statements in the text. Additionally what is the take-home message for the reader about the skewed distribution? Is it that the gains and losses are mostly small per site, except for a few sites where the gains and losses are very large? If so, please state this more clearly.
3. Line 148 says that 89.7% of the gained area was in the Atlantic Coast and line 149 says that 87.8% of the gained area was *Z. noltei*. Does this mean that the majority of the gained area was *Z. noltei* in the Atlantic Coast? If so, it would be good to state a percentage here for *Z. noltei* in the Atlantic Coast and perhaps elaborate a bit on this observation.
4. Lines 155-156 The symbol V will be unclear to readers who have not read the Methods first; I suggest changing this sentence so that knowledge of V is not necessary to understand this sentence.
5. Lines 158-160 this result appears to be primarily due to an increase in reporting; suggest a rephrase to avoid misinterpretation.
6. Lines 166-167 I wasn't sure what was the evidence for wasting disease being the dominant driver for losses of *Z. marina* in the Atlantic Coast because there isn't a figure, table or other reference here
7. Lines 169-170 I wasn't sure why improvement of water quality and regulating anchoring and trawling were the important management actions because the referenced Figure 5A and Table S2 don't mention these
8. Lines 171-12 I wasn't sure why recovery from wasting disease in the 1950s, recovery after drastic losses in coastal lagoons caused by floods, and colonisation were the important examples of natural recolonization because the referenced Table S2 doesn't mention these
9. Line 180 suggest rephrasing as "the maximum compiled area, due to several causes including wasting disease, water quality degradation..."
10. Line 183 "since the 1950s"
11. Line 185 "and were mostly due to the recovery"
12. Line 193-194 I could not see how this sentence was supported by Table S2
13. Line 196 suggest change "were" to "included" since several other loss reasons are shown in Table S2
14. Line 200 "For *Z. noltei*" (remove the word "As")
15. Line 204 reference Figure 3
16. Line 205 Replace "this decade" with "the 1950s" for clarity
17. Line 207 "*Zostera noltei* and *Z. marina* along the Atlantic Coasts" - see my comment #3
18. Line 209 "patent" - is this the wrong word?
19. "restoration of turbidity" ... should this be "restoration of water clarity"? And I am assuming the restoration of salinity involved changing the water from being more fresh water to more saline?
20. Line 231 "do not allow us to relate"

21. Lines 244-245 No need to repeat the words "Natura 2000" since the 2000 could be confused with a year, so suggest rephrasing to "which included 322 sites with *P. oceanica* meadows in the Mediterranean in 2006, encompassing"
22. Line 245 Change "Km" to "km"
23. Line 246 Change "complementarity" to "complementary"
24. Line 270 "520 potential" instead of "potential 520"
25. Line 289 "sites were in the Mediterranean Sea"
26. Lines 334-334 and 349-350 there is a repeated sentence, and I don't quite understand it
27. Line 340 "two observations closest to the decade boundaries"
28. Line 342-346 include a reference to Figure 4B
29. Line 370 "Restoration was never reported as a cause of seagrass gain." This is an interesting finding but I am concerned that it potentially conflicts with other literature. Bayraktarov et al. 2016 *Ecological Applications* 26: 1055-1074 "The cost and feasibility of marine coastal restoration" reviews several sites worldwide where seagrass restoration has been attempted, including in Europe, and appears to specifically refer to a successful seagrass restoration project in the Mediterranean (Balestri & Lardicci 2012 *Journal of Applied Ecology* 49: 1426-1435)? This point is important and needs elaboration in the Discussion.
30. Lines 371-373 I didn't understand these 2 sentences
31. Lines 376-377 Are there any places in the manuscript where mean plus or minus SE is used without specification? I couldn't see any. If not, this sentence can be deleted.
32. Figure 2 caption "the x represents" should be "the + represents"
33. Figure 5 caption needs to explain why half of the causes of decline listed in Figure 5A are not shown in Figure 5B. (Based on the previous version of the manuscript I think that wasting disease is not shown because it is outside of the time period of Figure 5B? But I am not sure what happened to the other causes of decline.)

**** Author's responses in red ****

REVIEWERS' COMMENTS:

Reviewer #2 (Remarks to the Author):

I applaud the authors of the manuscript "Recent trend reversal for declining European seagrass meadows" for their careful and thorough consideration of all reviewers' comments. The manuscript has significantly improved. I have some additional (minor) suggestions to improve the manuscript; mostly related to the changes they have made, after which I believe that the paper is suitable for publication in Nature Communications.

1. Abstract Line 68 If this statement is related to Figure 4B, it should be "the rate of change of seagrass density".

REPLY: this statement refers to both panels **a** (specific rate of change) and **b** (evolution of the trajectory) of Figure 4.

ACTION: including the name of the two variables is not possible due to the word restriction of the abstract. Thus, we replaced "seagrass density" by "density metrics" to use a general term for both variables (rate of change and evolution) (LINES 18-19).

2. Lines 126-139 I still find Figure S2 very unclear. I don't believe it helps to justify, or assist in the understanding of, the results mentioned in the text. I cannot understand how Figure S2 shows that 16 sites accounted for 75% of the losses and 5% sites accounted for the same percentage of gains (Lines 128-129). I also cannot understand how Figure S2 demonstrates a skewed distribution of the frequencies of losses and gains because the log scale on the x-axis hides the skew. Please consider another figure to justify these statements in the text. Additionally what is the take-home message for the reader about the skewed distribution? Is it that the gains and losses are mostly small per site, except for a few sites where the gains and losses are very large? If so, please state this more clearly.

REPLY: we agree that the interpretation of the main panel in Figure S2 (now Supplementary Figure 2) is not straightforward, and that it does not assist in the understanding of the results explained already in the text, so we decided to remove it.

Regarding the figure in the small panel, it sends two messages:

- 1) most of the changes in area (either losses or gains) are of the same size (ca. 1-100 ha for gains and ca. 10-1000 ha for losses; = central part of the distribution plot), and only a few are very large or very small (= tails of the distribution plots),
- 2) in comparison to the density plot of the gains, the distribution of the losses is displaced to the right by ca. one order of magnitude, meaning that losses are normally larger than gains (which is supported by the statistics shown in the main text).

We think that the lack of clarity of this figure was based on our misuse of the word "skewed" to explain the results. Since we believe that these messages are relevant to understand the scales of the gains and losses in seagrass area, we would like to keep the corresponding figure, and then explain better the information it provides in the main text.

ACTION: we deleted the main panel of the figure and kept only the small one (now re-sized) as Supplementary Figure 2. We rephrased the text in which the results associated to this figure are explained (LINES 94-97).

3. Line 148 says that 89.7% of the gained area was in the Atlantic Coast and line 149 says that 87.8% of the gained area was *Z. noltei*. Does this mean that the majority of the gained area was *Z. noltei* in the Atlantic Coast? If so, it would be good to state a percentage here for *Z. noltei* in the Atlantic Coast and perhaps elaborate a bit on this observation.

REPLY: yes, most of the area gains were observed for *Z. noltei* in the Atlantic coast. We agree that giving the percentages by species*region is more meaningful than as it is now.

ACTION: we have rewritten the results so now the contributions are given by combining the species and regions (LINES 105-108). This observation was already highlighted in the discussion (LINE 145, LINES 166-167).

4. Lines 155-156 The symbol V will be unclear to readers who have not read the Methods first; I suggest changing this sentence so that knowledge of V is not necessary to understand this sentence.

REPLY: " V " is the common symbol for the test statistic of the Wilcoxon signed rank test, a type of information that is compulsory to include following the Reporting Summary of Nature, when reporting the results of any statistical test.

ACTION: we have changed “V” by “statistic”, so the readers can understand that we are reporting the statistical results of the test.

5. Lines 158-160 this result appears to be primarily due to an increase in reporting; suggest a rephrase to avoid misinterpretation.

ACTION: we rephrased the sentence focussing on the comparison of the number of sites “worsening” and “improving” within each decade (which is the change observed from 1990s to 2000s) (LINES 114-120).

6. Lines 166-167 I wasn’t sure what was the evidence for wasting disease being the dominant driver for losses of *Z. marina* in the Atlantic Coast because there isn’t a figure, table or other reference here.

ACTION: we included a reference to Figure 5a, where the bar for wasting disease is only for *Z. marina*. We deleted “in the Atlantic coast” because the region is not shown in the figure (LINES 126-129).

7. Lines 169-170 I wasn’t sure why improvement of water quality and regulating anchoring and trawling were the important management actions because the referenced Figure 5A and Table S2 don’t mention these.

REPLY: the category “management actions” is defined in the Materials and Methods sections as “positive changes due to regulation and management, including removal/reduction of direct impacts such as improvement of water quality, trawling regulation, reduction of industrial sewage, anchoring regulation, and others”. We mentioned in the text “improvement of water quality and regulating anchoring and trawling” just to give some of those actions considered within the broad category “management actions”.

ACTION: in order to improve clarity: a) we moved the citation of figure 5a to an early position in the sentence so the reader will not expect to find those actions in the figure; b) we rephrased the second part of the sentence to clarify those actions were improvement of water quality, reduction of industrial sewage and the regulation of anchoring and trawling (LINES 128-131).

8. Lines 171-172 I wasn’t sure why recovery from wasting disease in the 1950s, recovery after drastic losses in coastal lagoons caused by floods, and colonisation were the important examples of natural recolonization because the referenced Table S2 doesn’t mention these.

REPLY: as for the previous comment, “recovery from wasting disease”, “recovery after drastic losses... and colonisation” are some of the sub-categories defined for “natural colonisation”, as defined in the Materials and Methods section.

ACTION: in order to improve clarity: a) we moved the citation of the Figure 5a to an earlier position in the sentence so the reader will not expect to find them in figure; b) we rephrased the second part of the sentence to clarify that those are examples of natural colonisation in the compilation (LINES 131-134). Note that table S2 was removed to meet editorial requirements, but the same info is in Figure 5a).

9. Line 180 suggest rephrasing as “the maximum compiled area, due to several causes including wasting disease, water quality degradation...”.

ACTION: we rephrased as suggested and added “combination of them [of the different causes]” (LINES 141-142).

10. Line 183 “since the 1950s”.

ACTION: done.

11. Line 185 “and were mostly due to the recovery”.

ACTION: done.

12. Line 193-194 I could not see how this sentence was supported by Table S2.

REPLY: in that sentence, we collated information from two sources: Table 1 for the species that lost the highest proportion of areas (57% for *Z. marina* and 46% for *C. nodosa*), and Table S2 for the causes of decline for those species. However, in the previous revision we decided to be more careful to correlate the causes and the losses because not always the compiled sites included both kind of information.

ACTION: We rephrased the sentence to clarify the species-specific losses and causes (LINES 154-156).

13. Line 196 suggest change “were” to “included” since several other loss reasons are shown in Table S2.

ACTION: done (LINE 156).

14. Line 200 “For *Z. noltei*” (remove the word “As”).

ACTION: done (LINE 160).

15. Line 204 reference Figure 3.

ACTION: done (LINE 164).

16. Line 205 Replace “this decade” with “the 1950s” for clarity.

ACTION: done (LINE 165).

17. Line 207 “Zostera noltei and Z. marina along the Atlantic Coasts” - see my comment #3

ACTION: we have specified the percentage of total gains due to each species, so the contribution of them are cleared now (LINES 166-168).

18. Line 209 “patent” – is this the wrong word?

ACTION: We changed “patent” by “evident” (= clearly seen) (LINE 169).

19. “restoration of turbidity” ... should this be “restoration of water clarity”? And I am assuming the restoration of salinity involved changing the water from being more fresh water to more saline?

REPLY: reviewer is right, should be “restoration of water clarity” and the salinity restoration implied an increased in salinity after the high inputs of freshwater due to the floods.

ACTION: we have rephrased the sentence as follows: “...due to the natural restoration of water clarity and salinity, which had been drastically reduced by two consecutive river floods” (LINES 174-176).

20. Line 231 “do not allow us to relate”.

ACTION: done (LINE 192).

21. Lines 244-245 No need to repeat the words “Natura 2000” since the 2000 could be confused with a year, so suggest rephrasing to “which included 322 sites with P. oceanica meadows in the Mediterranean in 2006, encompassing”.

ACTION: done (LINE 206).

22. Line 245 Change “Km” to “km”.

ACTION: done (LINE 206).

23. Line 246 Change “complementarity” to “complementary”.

ACTION: done (LINE 207).

24. Line 270 “520 potential” instead of “potential 520”.

ACTION: done (LINE 231).

25. Line 289 “sites were in the Mediterranean Sea”.

ACTION: done (LINE 249).

26. Lines 334-334 and 349-350 there is a repeated sentence, and I don't quite understand it.

REPLY: this sentence explained that only the rates of change for sites showing a change (either increase or decrease) were used in que quantitative analysis, for example to report average values (= Figure 4a).

ACTION: Since, Figure 4a is the only one that applies to this sentence, we added a brief note in the corresponding figure legend.

27. Line 340 “two observations closest to the decade boundaries”.

ACTION: done (LINE 300).

28. Line 342-346 include a reference to Figure 4B.

ACTION: done (LINE 306). Note Figure 4B is now Fig. 4d-f.

29. Line 370 “Restoration was never reported as a cause of seagrass gain.” This is an interesting finding but I am concerned that it potentially conflicts with other literature. Bayraktarov et al. 2016 Ecological Applications 26:1055-1074 “The cost and feasibility of marine coastal restoration” reviews several sites worldwide where seagrass restoration has been attempted, including in Europe, and appears to specifically refer to a successful seagrass restoration project in the Mediterranean (Balestri & Lardicci 2012 Journal of Applied Ecology 49:1426-1435)? This point is important and needs elaboration in the Discussion.

REPLY: we understand and share the reviewer's concern. Whereas it is true that some seagrass restoration trials have shown success in Europe, the most recent review on this topic (Cunha et al. 2012, Restoration Ecology, 20(4): 427-430) found that “none of the seagrass restoration programs developed in Europe by the participants during the last 10 years was successful. Furthermore, an informal review of data published in seagrass restoration success, showed that the results reported were biased because they were mostly based on a very short monitoring period (i.e. <1 year).”

In addition, the publications of successful seagrass restoration trials rarely report data of the increase in area or shoot density, but instead they normally report rate of success. For instance, in the study pointed out by the reviewer, Balestri & Lardicci 2012, they reported % survival and number of living shoots since the beginning of the restoration and for over a year (figure 4 in Balestri & Lardicci). Unfortunately, none of those were metrics in our compilation.

ACTION: to avoid conflicts with other literature and not to indirectly imply the absence of successful seagrass restoration projects in Europe, we clarified that “Restoration was not among the causes of seagrass gain in the compiled sources” (LINE 329-330).

30. Lines 371-373 I didn't understand these 2 sentences.

ACTION: the sentences were rephrased for clarity (LINES 330-335).

31. Lines 376-377 Are there any places in the manuscript where mean plus or minus SE is used without specification? I couldn't see any. If not, this sentence can be deleted.

ACTION: after checking that SE is always specified in the manuscript, we have deleted the sentence. Also, SE has been replaced by s.e.m. to follow editorial requests.

32. Figure 2 caption “the x represents” should be “the + represents”.

ACTION: done.

33. Figure 5 caption needs to explain why half of the causes of decline listed in Figure 5A are not shown in Figure 5B. (Based on the previous version of the manuscript I think that wasting disease is not shown because it is outside of the time period of Figure 5B? But I am not sure what happened to the other causes of decline.)

REPLY: there are 3 causes from Figure 5a not shown in Figure 5B (now Figure 5b-c): a) the cause “non-native macroalgae invasion”, because we had considered that there were not enough sites to create a “timeline” (n = 6, 2 in the 1990s and 4 in the 2000s); b) “wasting disease”, because, as the reviewer said, is outside the time period; and c) the “multiple causes”, because the counting of sites per decade was done for each cause even if the a site had more than one cause.

ACTION: a) We included “non-native macroalgae invasion” in Figure 5B (now panels **b-c**) for consistency, despite the low number of reports. b) We also included “wasting disease” in the legend of the figure 5B (now panels **b-c**) indicating that it was previous to 1950s so not depicted. c) We clarified in Materials and Methods that, when a site had multiple causes assigned, the site was counted for every cause.